# Jury-and-Judge Chain-of-Thought for Uncovering Toxic Data in 3D Visual Grounding

**Kaixiang Huang[1]    Qifeng Zhang[1]    Jin Wang[1]\*    Jingru Yang[2]**

**Yang Zhou[1]    Huan Yu[1]    Guodong Lu[1]    Shengfeng He[3]**
[1]Zhejiang University    [2]Carnegie Mellon University    [3]Singapore Management University

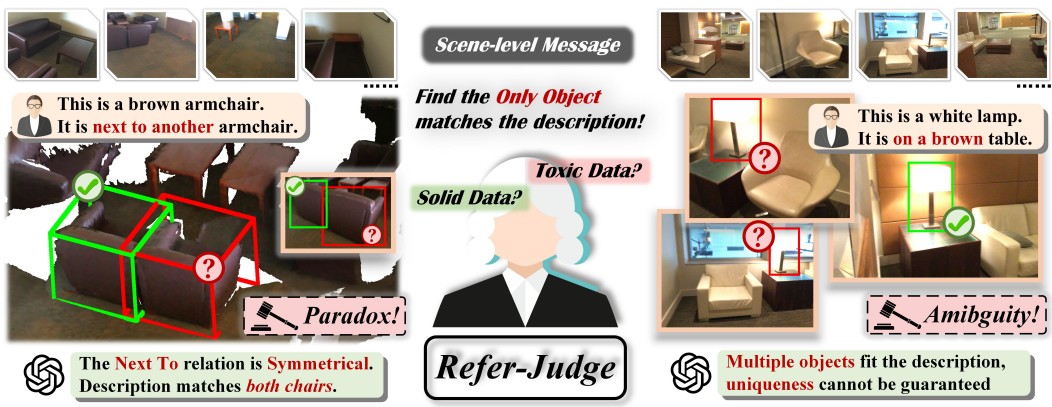

Figure 1: 3D visual grounding aims to localize a unique target object in a scene given a single referring expression, but existing datasets often contain ambiguous or paradoxical samples that hinder this goal. We propose *Refer-Judge*, an agentic, deliberation-based framework that detects and filters toxic data through structured multi-perspective reasoning and corroborative refinement.

## Abstract

3D Visual Grounding (3DVG) faces persistent challenges due to coarse scene-level observations and logically inconsistent annotations, which introduce ambiguities that compromise data quality and hinder effective model supervision. To address these challenges, we introduce *Refer-Judge*, a novel framework that harnesses the reasoning capabilities of Multimodal Large Language Models (MLLMs) to identify and mitigate toxic data. At the core of Refer-Judge is a *Jury-and-Judge Chain-of-Thought* paradigm, inspired by the deliberative process of the judicial system. This framework targets the root causes of annotation noise: jurors collaboratively assess 3DVG samples from diverse perspectives, providing structured, multi-faceted evaluations. Judges then consolidate these insights using a *Corroborative Refinement* strategy, which adaptively reorganizes information to correct ambiguities arising from biased or incomplete observations. Through this two-stage deliberation, Refer-Judge significantly enhances the reliability of data judgments. Extensive experiments demonstrate that our framework not only achieves human-level discrimination at the scene level but also improves the performance of baseline algorithms via data purification. Code is available at `https://github.com/Hermione-HKX/Refer_Judge`.

---

\*Corresponding author. `dwjcom@zju.edu.cn`

# 1 Introduction

3D Visual Grounding (3DVG) aims to localize a unique target object within a complex 3D scene based on a natural language referring expression. As a critical bridge between human perception and machine understanding of real-world environments, it has driven the development of large-scale datasets such as the widely adopted ScanRefer [7], laying the groundwork for algorithmic advances.

To support generalizable learning, 3DVG datasets must capture diverse scenes and densely annotate referable objects. However, meeting this demand places a heavy burden on annotators, who must repeatedly write descriptions while navigating incomplete and often ambiguous scene representations—namely, sparse 3D point clouds and disjointed 2D frames. This process has led to a non-negligible number of *toxic samples*, whose descriptions degrade training quality and compromise evaluation reliability. These toxic samples typically arise from two sources: 1) *Logical paradoxes*, where the description is internally inconsistent and fails to isolate a single target (e.g., *"... next to another ..."* creates symmetrical ambiguity); and 2) *Referential ambiguities*, where vague or under-specified descriptions match multiple similar objects. Figure 1 illustrates these cases: the left side shows a logically paradoxical reference, while the right highlights under-specification due to insufficient scene observation.

Detecting such toxic data requires deep, multimodal reasoning across both 3D scene structure and natural language. Yet large-scale human validation is costly, time-consuming, and difficult to scale [26, 54]. Recent advances in Multimodal Large Language Models (MLLMs) [1, 31, 33, 13, 20] have shown remarkable capabilities in perception and reasoning [14, 35], opening up new possibilities for data evaluation. For example, SeeGround [19] enhances MLLM-based understanding using adaptive 2D renderings and achieves competitive zero-shot performance on 3DVG tasks. However, current research largely treats MLLMs as perceptual engines or grounding backbones [49, 15, 16, 9], overlooking their potential to identify hidden toxic samples. This gap highlights a critical yet underexplored direction: employing MLLMs for principled data quality assessment in 3DVG.

Meanwhile, MLLMs are increasingly recognized as capable evaluators. JudgeLM [55], for instance, fine-tunes LLMs to assess output reliability, while others [50, 6, 45, 52] explore Chain-of-Thought (CoT) prompting [37] to improve alignment with human judgment. Despite encouraging results, these approaches are generally limited to pairwise comparisons or simple content scoring, and they falter when faced with the structural complexity of scene-level reasoning. Successors like CoCoT [45] and GoT [5] target fragmented, open-domain tasks (e.g., single questions or isolated image-text pairs) and are ill-equipped to trace the nuanced origins of toxic data in 3DVG. The lack of a targeted, coherent judgment mechanism capable of deep multimodal reasoning leaves toxic sample detection in 3DVG an unsolved problem.

To address this challenge, we introduce **Refer-Judge**, a novel agentic deliberation-based system for uncovering toxic annotations in 3DVG. Unlike prior MLLM evaluators, Refer-Judge adopts a *Jury-and-Judge Chain-of-Thought* paradigm inspired by judicial deliberation. Given a scene and its corresponding referring expression, multiple *Jurors* analyze the sample from four core perspectives: *Logic*, *Consistency*, *Distinguishability*, and *Ambiguity*. This structured division of reasoning enables jurors to identify logical contradictions and ambiguities from diverse angles, forming a comprehensive diagnosis of sample quality.

Mirroring the real-world jury system, this division of reasoning promotes both coverage and specialization in evaluating heterogeneous scene-level inputs. The outcomes from the jurors are then passed to a panel of *Magistrate Judges*, each responsible for consolidating feedback within a specific dimension, ensuring internal coherence and resolving intra-aspect contradictions. To synthesize a final verdict, one *District Judge* performs high-level arbitration through targeted re-evaluation. Central to this stage is our proposed *Corroborative Refinement* strategy, which goes beyond majority voting or self-consistency heuristics [34, 5]. Instead of relying on static consensus, Magistrate Judges adaptively reorganize visual and textual evidence based on juror insights, retrieving auxiliary context and correcting uncertain reasoning caused by blurred or incomplete observations. This process yields judgments that are both context-aware and resilient to scene complexity. By integrating multi-role, multi-perspective reasoning without reliance on task-specific fine-tuning or external perception modules (e.g., object detectors or renderers), Refer-Judge delivers robust scene-level assessments. Extensive experiments demonstrate that Refer-Judge achieves human-comparable performance in identifying toxic samples, enhancing the quality of supervision, and advancing the reliability of 3DVG benchmarks.

In summary, our main contributions are threefold:

- We identify and systematically analyze the prevalence of paradoxical and ambiguous samples in existing 3DVG datasets, and present Refer-Judge, a novel evaluation method capable of uncovering toxic annotations without reliance on task-specific fine-tuning or auxiliary perception modules.

- We propose a Jury-and-Judge Chain-of-Thought approach, wherein jurors perform targeted, multi-perspective reasoning and a panel of judges, culminating in high-level refinement, delivers robust, structured judgments through our Corroborative Refinement mechanism.

- We validate the effectiveness of Refer-Judge through extensive experiments, showing both its alignment with human judgments and its ability to improve baseline 3DVG model performance when trained on filtered data.

## 2 Related Work

**3D Visual Grounding and Multimodal Large Language Models.** 3DVG focuses on localizing a unique target object in a 3D scene based on a natural language referring expression. Early approaches primarily adopt a detection-then-match pipeline, emphasizing improved alignment between visual and linguistic modalities [7, 22, 51, 46, 47, 36, 29]. With the rapid advancement of Large Language Models (LLMs)[1, 18, 31, 13, 20], recent efforts have explored the potential of MLLMs for 3DVG tasks[49, 19, 28, 15, 11]. For instance, 3D-LLM [15] integrates point cloud representations into LLMs and shows strong performance across diverse 3D reasoning tasks. SeeGround [19] leverages dynamically rendered 2D views to enhance open-vocabulary grounding, achieving competitive zero-shot results. And CoT3DRef [4] further introduce the idea of CoT into 3DVG to form a more interpretable Seq2Seq staged prediction. However, while these approaches expand the capability of MLLMs, they largely overlook the quality of the underlying datasets. Paradoxical and ambiguous annotations, often caused by human error or limited scene observability, remain unaddressed and continue to degrade both model training and evaluation reliability.

**LLMs as Evaluators.** A growing body of work explores the use of LLMs as evaluators, often referred to as *LLM-as-a-Judge*, to approximate human judgment in various domains [14, 43, 3, 55, 54]. For example, MLLM-as-a-Judge [6] evaluates MLLMs' decision-making capabilities across tasks and highlights their limitations in handling fine-grained reasoning. FINCON [43] applies a multi-agent LLM setup to assess financial risk, while LLM-Grounder [40] uses MLLMs to score candidate views in 3DVG, enhancing zero-shot grounding performance. Although these studies demonstrate the promise of LLMs as evaluators, most are restricted to shallow or fragmented inputs, such as isolated prompts or image-text pairs. The use of MLLMs for structured, scene-level evaluation in complex multimodal tasks like 3DVG remains relatively unexplored.

**Chain-of-Thought Reasoning.** Chain-of-Thought (CoT) prompting enhances LLM reasoning by encouraging step-by-step inference [37]. Initial methods such as vanilla CoT introduce sequential logic to improve answer reliability. More advanced techniques like Tree-of-Thought (ToT)[42, 21], Graph-of-Thought (GoT)[5], and Hierarchy-of-Thought (HoT)[41] enable models to explore multiple reasoning paths or build graph-based reasoning structures. In multimodal settings, extensions such as G-CoT[23] and MC-CoT [38] combine visual and textual modalities for more comprehensive analysis. Despite these advances, most CoT-based methods are tailored for narrow-scope tasks, including arithmetic reasoning or short image-text question answering [53, 48, 45, 50, 52]. Their refinement strategies generally involve simple path selection, answer re-ranking, or majority voting [5, 34, 30, 27, 24]. These mechanisms are insufficient for evaluating rich, scene-level content where ambiguity stems from complex visual and linguistic interplay. In contrast, our proposed **Refer-Judge** enables distributed reasoning across multiple perspectives and introduces corroborative refinement for robust scene-level judgment in 3DVG.

## 3 Methods

To address the issue of toxic samples in 3D visual grounding (3DVG) datasets, we propose **Refer-Judge**, a novel *Jury-and-Judge Chain-of-Thought* framework that enables structured, scene-level reasoning over multimodal inputs. We begin by formulating the *LLM-as-a-Judge* task in the context of 3DVG data exploration and present an overview of our framework in Section 3.1. Section 3.2 details our Jury-based distributed evaluation strategy, which decomposes the assessment into multiple

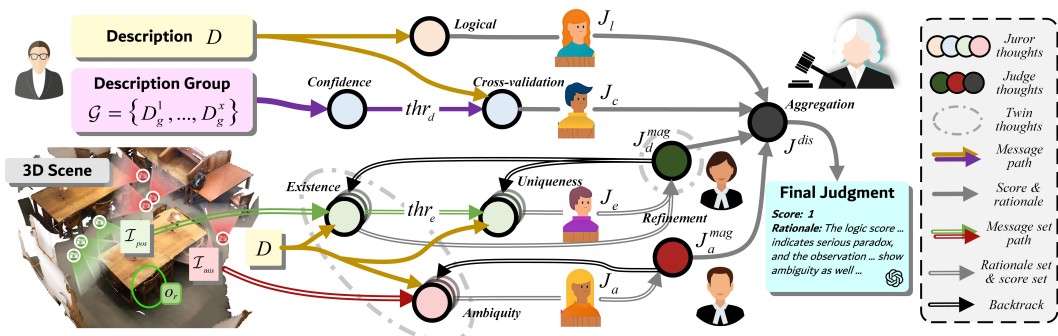

Figure 2: Overview of Refer-Judge. Jurors evaluate the scene and referring expression across four complementary dimensions. Their judgments are then refined and integrated to generate a final scene-level judgment.

aspects. Section 3.3 describes how the Judge module refines and integrates the reasoning paths to produce a final judgment.

## 3.1 Problem formulation and method overview

Consider an annotated 3D visual grounding (3DVG) dataset consisting of a 3D scene, where a set of images $\mathcal{I}$ from multiple viewpoints represents different regions of the environment, and a natural language description $D$ refers to a specific target object $o_r$. Our ultimate goal is to construct an agentic, deliberation-based judgment system $f_{MLLM}(\mathcal{I}, D)$ that can robustly and effectively identify toxic data caused by paradoxes and ambiguities. The system outputs discrete quality scores to reflect the reliability of each data sample, enabling a clear and interpretable evaluation of 3DVG data quality [3].

In contrast to prevailing LLM-based evaluators that focus solely on analyzing text-image pairs with limited complexity, the proposed Refer-Judge approach towards multifaceted CoT analysis for the scene-level multimodal information, thereby enabling a comprehensive data judgment. As demonstrated in Figure 2, jurors $\{J_l, J_c, J_d, J_a\}$ in Jury $\mathcal{J}_{jury}$ are employed in parallel to access 3DVG data from four key perspectives: *logical*, *consistency*, *distinguishability*, and *ambiguity*. To further enhance judgment reliability, the Judge system $\mathcal{J}_{judge}$ applies two Magistrate Judges, $J_d^{mag}$ and $J_a^{mag}$, tasked with aggregating and refining the outputs of the corresponding jurors across multiple image-description pairs. These judges adaptively identify uncertain or conflicting sub-evaluations and reorganize them by referencing more consistent information, thereby mitigating errors caused by limited viewpoints or incomplete observations. Finally, a District Judge $J^{dis}$ integrates all streams of reasoning and adjudicates the final quality score. Notably, due to Refer-Judge involving multifaceted assessment, we observed an interesting CoT phenomenon, where the same thought can be shared across different inputs and serve distinct judgment processes. We term this phenomenon *twin thought*, as marked in Figure 2.

## 3.2 Jury Reasoning

Aligned with the common causes of annotation errors in 3DVG datasets, the Refer-Judge initially introduces four jurors to comprehensively evaluate the heterogeneous scene-level information, as shown in Figure 3. Specifically, the textual modality focuses on identifying potential logical paradoxes within the referring expression, while the visual modality then uncovers ambiguity arising from incomplete or biased observations.

**Logical Juror.** The natural language referring expression is central to identifying the target object within the scene. To prevent self-contradictory or logically invalid descriptions, we introduce a logical Juror $J_l$ to assess the logical soundness of the input description. Given that logical reasoning is a fundamental ability of large language models, and 3DVG descriptions are typically short and simple, we adopt a direct strategy based on in-context learning to generate both a judgment score $A_l$ and rationale $R_l$, formulated as:

$$(A_l, R_l) = J_l(P_l, D) \tag{1}$$

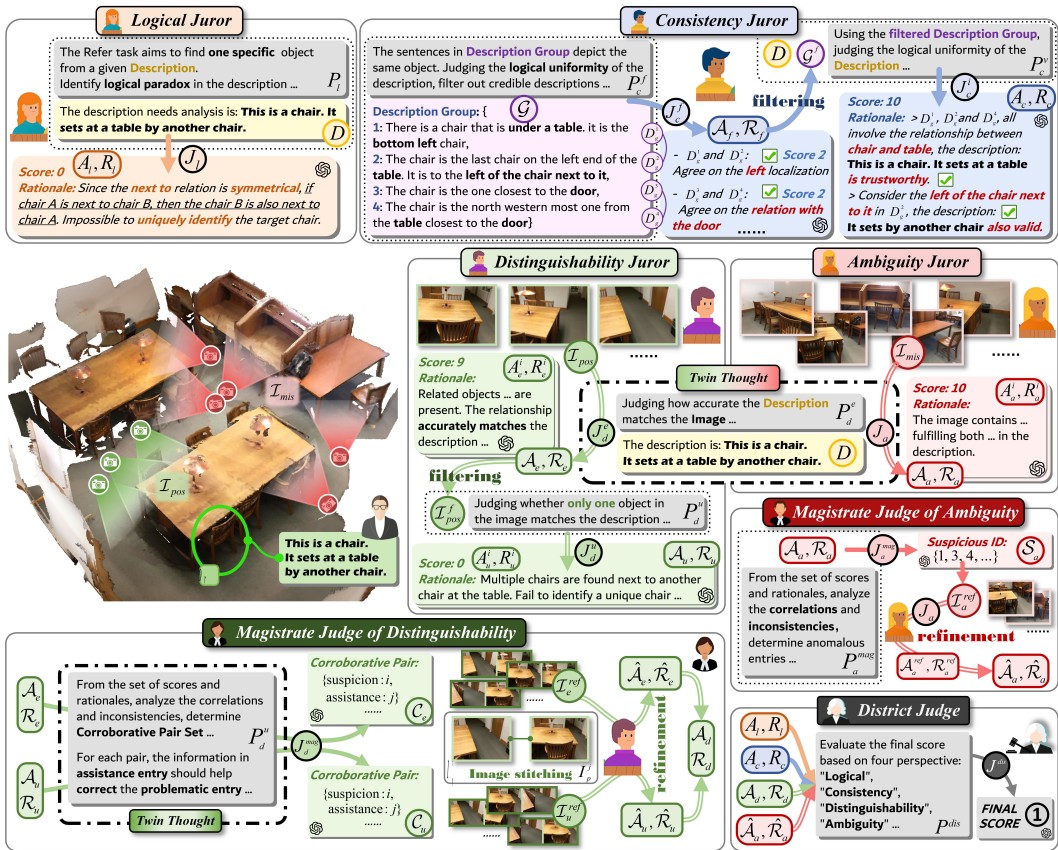

Figure 3: The detailed architecture of Refer-Judge. Submodules, distinguished by color, illustrate the hierarchical reasoning structure and information flow from parallel analysis to final decision-making within the Jury-and-Judge Chain-of-Thought.

where $P_l$ is the task-specific prompt provided to the juror $J_l$.

**Consistency Juror.** Beyond assessing the target description $D$ in isolation, we further introduce a consistency evaluation that leverages alternative descriptions of the same object across the dataset [7], denoted as the *Description Group* $\mathcal{G} = \{D_g^1, ..., D_g^x\}$. These descriptions serve as circumstantial evidence to evaluate the consistency of $D$, helping to prevent potential paradoxes across data entries.

The Consistency Juror $J_c$ is designed as a two-step chain-of-thought process. Since expressions within $\mathcal{G}$ may themselves be logically flawed, the first filtering step $J_c^f$ evaluates the internal consistency of this group. Removing descriptions with low confidence scores $A_f^i$ through a threshold $thr_d$, resulting in a refined subset $\mathcal{G}^f$. Next, $J_c^v$ performs cross-validation between the description $D$ and $\mathcal{G}^f$, producing the consistency score $A_c$ and rationale $R_c$. The process is formulated as:

$$(\mathcal{A}_f, \mathcal{R}_f) = J_c^f(P_c^f, \mathcal{G}), \quad \mathcal{G}^f = \{D_g^i \mid i \in [1, x], A_f^i \in \mathcal{A}_f, A_f^i > thr_d\}$$
$$(A_c, R_c) = J_c^v(P_c^v, \mathcal{G}^f, D) \tag{2}$$

where $P_c^f$ and $P_c^v$ denote the prompts used by $J_c^f$ and $J_c^v$, respectively. $\mathcal{A}_f$ and $\mathcal{R}_f$ are the confidence score and rationale sets for each candidate sentence in the description group.

**Distinguishability Juror.** Among all scene-level images $\mathcal{I}$, the subset of views that contain the referred object $o_r$, denoted as $\mathcal{I}_{pos} = \{I_p^1, ..., I_p^x\} \subseteq \mathcal{I}$, serves as the positive set essential for assessing distinguishability. Specifically, we align the RGB frames with 3D annotations to identify images where $o_r$ is visible, ensuring that all views in $\mathcal{I}_{pos}$ depict the intended referent. The verification process by human experts further involves: 1) checking whether the object described appears in the image, and 2) confirming that the expression refers to a unique object.

Mimicking human observation, the Distinguishability Juror $J_d$ also applies a two-stage CoT, jointly evaluating *existence* and *uniqueness* to derive a comprehensive distinguishability assessment. Initially, $J_d^e$ traverses each image-description pair, resulting in a set of existence scores $\mathcal{A}_e$ and rationales $\mathcal{R}_e$. Based on $thr_e$, $J_d$ filter out a subset $\mathcal{I}_{pos}^f$ that meets the conditions. Subsequently, $J_d^u$ further evaluates whether each remaining image contains a unique object satisfying the description, producing uniqueness scores $\mathcal{A}_u$ and rationales $\mathcal{R}_u$. In summary, the reasoning chain is expressed as follows:

$$(\mathcal{A}_e, \mathcal{R}_e) = \left\{ J_d^e(P_d^e, I_p^i, D) \mid I_p^i \in \mathcal{I}_{pos} \right\}, \quad \mathcal{I}_{pos}^f = \left\{ I_p^i \mid i \in [1, x], A_e^i \in \mathcal{A}_e, A_e^i > thr_e \right\}$$
$$(\mathcal{A}_u, \mathcal{R}_u) = \left\{ J_d^u(P_d^u, I_p^i, D) \mid I_p^i \in \mathcal{I}_{pos}^f \right\} \tag{3}$$

where $P_d^e$ and $P_d^u$ denote the used prompts.

**Ambiguity Juror.** While $J_d$ focuses on confirming object identification within positive views, the Ambiguity Juror $J_a$ provides a complementary perspective by evaluating whether other similar objects in the scene also satisfy the description, revealing potential ambiguity.

Notably, we introduce the concept of *twin thought*: under identical prompting conditions used in existence analysis, altering the input to achieve a distinct evaluation purpose. Specifically, $J_a$ applies the same existence assessment as $J_d^e$, but on a misleading view set $\mathcal{I}_{mis} = \left\{ I_m^1, ..., I_m^y \right\} \subseteq \mathcal{I}$, the misleading set is formed by selecting images from the same scene that do not contain the ground-truth object but include other instances of the same object category of $o_r$. If high confidence is assigned to $\mathcal{I}_{mis}$, it contradicts the exclusivity requirement of the 3DVG task, as follows:

$$(\mathcal{A}_a, \mathcal{R}_a) = \left\{ J_a(P_d^e, I_m^i, D) \mid I_m^i \in \mathcal{I}_{mis} \right\} \tag{4}$$

where $\mathcal{A}_a$ and $\mathcal{R}_a$ represent the sets of resulting scores and rationales, which serve as strong indicators of potential ambiguity.

## 3.3 Judge Refinement and Aggregation

As shown in Figure 3, to consolidate the distributed reasoning paths, Refer-Judge then introduces the Judge system $\mathcal{J}_{judge}$ introduces Magistrate Judges for refinement, followed by a District Judge that aggregates all juror outputs from multi-dimensional and heterogeneous assessment to produce a final, holistic quality scene-level judgment.

**Magistrate Judge of Distinguishability.** Due to incomplete or occluded views of the target object, both steps of the Distinguishability Juror $J_d$ may produce unreliable assessments. Therefore, identifying and refining flawed evaluations becomes a natural necessity. Unlike existing CoT refinement approaches based on majority voting or self-ranking [34, 5], the proposed Magistrate Judge of Distinguishability $J_d^{mag}$ moves beyond the simple selection and optimization of static consensus. Instead, it introduces a *Corroborative Refinement* mechanism, which focuses on retrieving auxiliary information and reorganizing inputs to enable more reliable re-evaluation.

Take the refinement of existence assessment as an example. The magistrate first examines the previous outputs $(\mathcal{A}_e, \mathcal{R}_e)$ and constructs a *Corroborative Pair Set* $\mathcal{C}_e$ by analyzing the associations and conflicts within the evaluation set. Each element (suspicion, assistance) corresponds to a pair of image indices $(i, j)$ from $\mathcal{I}_{pos}$, where the suspicion sample $I_p^i$ is considered potentially unreliable, and the corresponding assistance sample $I_p^j$ provides visual evidence to support re-evaluation (e.g., combining a complete view with an occluded one). Through stitching operation $sti(\cdot)$ and re-assessment via $J_d^e$, the refined existence outputs $\left(\mathcal{A}_e^{ref}, \mathcal{R}_e^{ref}\right)$ replace the originals, mitigating the effects of unreliable observations. Concretely, $sti(\cdot)$ horizontally concatenates the aligned edges of suspicion and assistance views to form a richer, context-aware visual prompt for final assessment.

On the other hand, the refinement of uniqueness judgment serves as the *twin thought* of existence refinement, performing a similar analysis of the corroborative pair $\mathcal{C}_u$, reorganization of visual information, and re-evaluation. The overall procedure is summarized as:

$$\mathcal{C}_{\{e,u\}} = J_d^{mag}(P_d^{mag}, \mathcal{A}_{\{e,u\}}, \mathcal{R}_{\{e,u\}}), \quad \mathcal{I}_{\{e,u\}}^{ref} = \left\{ sti(I_p^i, I_p^j) \mid (i, j) \in \mathcal{C}_{\{e,u\}} \right\}$$
$$\left( \mathcal{A}_{\{e,u\}}^{ref}, \mathcal{R}_{\{e,u\}}^{ref} \right) = \left\{ J_d^{\{e,u\}}(P_d^{\{e,u\}}, I, D) \mid I \in \mathcal{I}_{\{e,u\}}^{ref} \right\}$$
$$\left( \widehat{\mathcal{A}}_{\{e,u\}}, \widehat{\mathcal{R}}_{\{e,u\}} \right) = replace((\mathcal{A}_{\{e,u\}}, \mathcal{R}_{\{e,u\}}), (\mathcal{A}_{\{e,u\}}^{ref}, \mathcal{R}_{\{e,u\}}^{ref})) \tag{5}$$
$$(\mathcal{A}_d, \mathcal{R}_d) = (\widehat{\mathcal{A}}_e \oplus \widehat{\mathcal{A}}_u, \ \widehat{\mathcal{R}}_e \oplus \widehat{\mathcal{R}}_u)$$

where $P_d^{mag}$ denotes the prompt used for identifying suspicious-assistant pairs, $\mathcal{A}_d$ and $\mathcal{R}_d$ denote the final distinguishability score set and rationale set, with $\oplus$ representing the message combination. Meanwhile, $replace(\cdot)$ substitutes the original (possibly erroneous or low-confidence) score and rationale produced by individual jurors with the refined outputs, ensuring more accurate and globally consistent results.

**Magistrate Judge of Ambiguity.** Unlike the Distinguishability judgment, which operates over $\mathcal{I}_{pos}$ focused on the target object, the ambiguity assessment involves scattered distractor views with limited inter-image correlation. Therefore, the Magistrate Judge of Ambiguity $J_a^{mag}$ simplifies the refinement process by discarding corroborative pair analysis and directly identifying a set of suspicious assessments $\mathcal{S}_a$. As a simplified version to $J_d^{mag}$, the procedure is defined as:

$$
\begin{aligned}
\mathcal{S}_a &= J_a^{mag}(P_a^{mag}, \mathcal{A}_a, \mathcal{R}_a), \quad \mathcal{I}_a^{ref} = \left\{ I_m^i \mid i \in \mathcal{S}_a \right\} \\
\left( \mathcal{A}_a^{ref}, \mathcal{R}_a^{ref} \right) &= \left\{ J_a(P_d^e, I, D) \mid I \in \mathcal{I}_a^{ref} \right\} \\
\left( \widehat{\mathcal{A}}_a, \widehat{\mathcal{R}}_a \right) &= replace \left( (\mathcal{A}_a, \mathcal{R}_a),\ (\mathcal{A}_a^{ref}, \mathcal{R}_a^{ref}) \right)
\end{aligned}
\tag{6}
$$

**District Judge.** After distributed evaluations and localized refinements, the final decision is rendered by the District Judge $J^{dis}$, which integrates the reasoning paths provided by all preceding components. As an aggregation thought module, $J^{dis}$ focuses solely on analyzing the outputs of the Jurors and Magistrate Judges, without directly revisiting the original scene-level inputs $(\mathcal{I}, D)$. This design mirrors the real-world *Jury-and-Judge system*, preserving interpretability while avoiding interference from redundant original messages. The process is as follows:

$$
(A_j, R_j) = J^{dis} \left( P^{dis}, (A_l, R_l), (A_c, R_c), (\mathcal{A}_d, \mathcal{R}_d), \left( \widehat{\mathcal{A}}_a, \widehat{\mathcal{R}}_a \right) \right)
\tag{7}
$$

where $P^{dis}$ is the prompt guiding the final judgment, and $(A_j, R_j)$ denote the final judgment score with rationale. Altogether, the proposed Refer-Judge represent a comprehensive and trustworthy assessment of the 3DVG data, effectively uncovering latent paradoxes and ambiguities.

# 4 Experiments

We evaluate the performance of the proposed Refer-Judge as follows: Section 4.1 describes the experimental setup and implementation details; Section 4.2 presents the main results of Refer-Judge as well as the performance of baseline models after toxic data removal via Refer-Judge; Section 4.3 provides a detailed analysis of ablation studies on the proposed methods.

## 4.1 Setups and Implementation Details

**Datasets.** We conduct experiments on the proposed *ScanRefer-Justice* dataset to verify the effectiveness of Refer-Judge. Built upon the widely-used *ScanRefer* benchmark [7], ScanRefer-Justice introduces reliable 3DVG judgments annotated and verified by human experts, while emphasizing complex scenes with numerous similar objects, providing an accurate method assessment. Detailed statistics and construction procedures are provided in Appendix A. Additionally, we evaluate baseline models on the ScanRefer dataset to demonstrate how identifying and removing toxic annotations improves model performance.

**Baselines.** To validate the generality and effectiveness of Refer-Judge, we apply it across various representative Multimodal Large Language Models (MLLMs), including GPT-4o [18], GPT-4.1-mini [25], Grok-3 [39], Gemini-2.5 Pro [12], and LLaMA-3.2 11B [13]. To assess the impact of data purification on 3DVG, we consider multiple baseline models: *ScanRefer* [7], the pioneer method in 3DVG that established the detect-then-match paradigm; *3DVLP* [47] and ConcreteNet [32], which reflects superior performance in 3DVG and multiple downstream tasks.

**Evaluation Metrics.** Following JudgeLM [55], we apply agreement rate, precision, recall, and F1-score to verify the effect of toxic data identification. The RMSE and MAE are also reported to better quantify the alignment with human judgment on fine-grained scoring. To evaluate 3DVG performance, following the experiment setting in [7], we choose Acc@0.25 and Acc@0.5 as the evaluation metrics. A more precise definition of the evaluation metric is provided in Appendix B.

Table 1: Quantitative results on ScanRefer-Justice dataset. The **Bold** denotes the best performance. The *Human Performance* is statistically obtained during the verification process of the annotation.

| Model | Agreement ↑ | Precision ↑ | Recall ↑ | F1 ↑ | RMSE ↓ | MAE ↓ |
|---|---|---|---|---|---|---|
| GPT-4o [18] | **82.77** | **82.95** | **85.77** | **84.33** | **2.69** | **1.71** |
| GPT-4.1-mini [25] | 81.81 | 82.64 | 83.66 | 83.14 | 2.82 | 1.94 |
| Grok-3 [39] | 81.14 | 81.03 | 84.66 | 82.81 | 3.07 | 1.84 |
| Gemini-2.5 Pro [12] | 77.01 | 78.53 | 78.39 | 78.53 | 3.15 | 2.20 |
| LLAMA-3.2-11B [13] | 67.88 | 67.67 | 76.83 | 71.96 | 3.71 | 2.73 |
| *Human Performance* | 84.87 | 90.43 | 82.92 | 86.51 | - | - |

Table 2: Quantitative results on ScanRefer dataset. (a) shows results under the original dataset splitting sets. (b) shows results under separated toxic and purified samples.

(a) Quantitative results on ScanRefer dataset. The models are verified on the original validation set.

| Method | Unique ↑ | | Multiple ↑ | | Overall ↑ | |
|---|---|---|---|---|---|---|
| | Acc@0.25 | Acc@0.5 | Acc@0.25 | Acc@0.5 | Acc@0.25 | Acc@0.5 |
| TGNN [17] | 68.61 | 56.80 | 29.84 | 23.18 | 37.37 | 29.70 |
| InstanceRefer [44] | 75.72 | 64.66 | 29.41 | 22.99 | 38.40 | 31.08 |
| 3DVG-Transformer [51] | 81.93 | 60.64 | 39.30 | 28.42 | 47.57 | 34.67 |
| SeeGround [19] | 75.7 | 68.9 | 34.0 | 30.0 | 44.1 | 39.4 |
| 3D-VisTA [56] | 81.6 | 75.1 | 43.7 | 39.1 | 50.6 | 45.8 |
| ScanRefer [7] | 76.33 | 53.51 | 32.73 | 21.11 | 41.19 | 27.40 |
| + *Refer-Judge* | 79.57(+3.24) | 54.31(+0.8) | 34.15(+1.42) | 22.69(+1.58) | 42.96(+1.77) | 28.83(+1.43) |
| 3DVLP [47] | 85.18 | 70.04 | 43.65 | 33.40 | 51.70 | 40.51 |
| + *Refer-Judge* | 86.29(+1.11) | 72.19(+2.15) | 44.24(+0.59) | 34.88(+1.48) | 52.39(+0.69) | 42.11(+1.60) |
| ConcreteNet [32] | 82.39 | 75.62 | 41.24 | 36.56 | 48.91 | 43.84 |
| + *Refer-Judge* | 84.14(+1.75) | 79.57(+3.95) | 41.97(+0.73) | 36.16(-0.40) | 49.94(+1.03) | 44.55(+0.71) |

(b) Quantitative results on toxic and purified data of ScanRefer val set.

| Method | Toxic data ↓ | | Unique (purified) ↑ | | Multiple (purified) ↑ | | Overall (purified) ↑ | |
|---|---|---|---|---|---|---|---|---|
| | Acc@0.25 | Acc@0.5 | Acc@0.25 | Acc@0.5 | Acc@0.25 | Acc@0.5 | Acc@0.25 | Acc@0.5 |
| ScanRefer | 21.69 | 14.58 | 76.89 | 50.57 | 34.96 | 21.80 | 43.91 | 27.94 |
| + *Refer-Judge* | 18.76(-2.93) | 13.67(-0.91) | 79.50(+2.61) | 55.41(+4.84) | 36.07(+1.11) | 24.44(+2.64) | 45.33(+1.42) | 31.04(+3.10) |
| 3DVLP | 22.91 | 17.75 | 85.97 | 70.02 | 46.44 | 36.02 | 54.86 | 43.01 |
| + *Refer-Judge* | 22.11(-0.80) | 15.46(-2.29) | 86.41(+0.44) | 72.27(+2.25) | 47.14(+0.70) | 37.43(+1.41) | 55.50(+0.64) | 44.85(+1.84) |
| ConcreteNet | 26.33 | 22.37 | 82.73 | 75.68 | 42.32 | 37.42 | 50.92 | 45.24 |
| + *Refer-Judge* | 24.18(-2.15) | 21.92(-1.08) | 84.03(+1.30) | 79.23(+3.55) | 43.03(+0.71) | 37.97(+0.55) | 51.44(+0.52) | 46.45(+1.21) |

**Implementation Details.** To ensure generality, all MLLMs are used with default configurations. Proprietary models, such as GPT-4o, are accessed through APIs. LLAMA-3.2 is deployed with the released checkpoints. Our experiments are conducted on a computational platform equipped with Intel(R) Xeon(R) CPU E5- 2680v3 @2.50 GHz CPU x2, 128G memory, and RTX 4090 GPU x8. The inference of LLAMA is conducted using a dual-GPU setup.

## 4.2 Main Results

**Comparison on ScanRefer-Justice Benchmark.** We first evaluate the proposed Refer-Judge approach on the ScanRefer-Justice datasets. As shown in Table 1, the quantitative results cover the performance instantiated with various MLLMs to comprehensively assess its effectiveness and adaptability. Notably, all models are evaluated in a zero-shot manner without fine-tuning on ScanRefer-Justice. Specifically, the results reveal a strong correlation between the identification of toxic data and the underlying reasoning and perception capabilities of backbones. Among them, Refer-Judge (GPT-4o) achieves human-level judgment capability, slightly lagging with human experts by -2.1% in agreement score, while achieving a recall of over 85%. The subsequent experiments are all based on the GPT-4o-driven Refer-Judge. Additional case studies are presented in Appendix D.

**Comparison on ScanRefer Benchmark.** We further investigate the impact of applying Refer-Judge to identify and filter toxic data from the ScanRefer training set. As shown in Table 2a, all baseline

Table 3: Ablation of the Jury System. **Bold** denotes best performance.

| Method | Agmt. ↑ | Prec. ↑ | Rec. ↑ | F1 ↑ |
|---|---|---|---|---|
| Jury system (Ours) | **82.77** | **82.95** | **85.77** | **84.33** |
| Mix logical | 80.52 | 80.34 | 84.70 | 82.46 |
| Mix observation | 76.60 | 77.15 | 80.21 | 78.65 |
| Mix four branches | 75.67 | 76.03 | 79.91 | 77.91 |

Table 4: Ablation of the Judge System. **Bold** denotes best performance.

| Method | Agmt. ↑ | Prec. ↑ | Rec. ↑ | F1 ↑ | RMSE ↓ | MAE ↓ |
|---|---|---|---|---|---|---|
| CR (Ours) | **82.77** | 82.95 | **85.77** | **84.33** | 2.69 | **1.71** |
| CoT-SC [34] | 82.54 | **83.17** | 84.61 | 83.28 | 2.81 | 1.91 |
| GoT [5] | 81.55 | 82.73 | 83.25 | 82.99 | **2.66** | 1.78 |
| Average | 74.24 | 74.63 | 78.85 | 76.68 | 3.27 | 2.21 |
| w/o refinement | 81.72 | 83.15 | 83.00 | 83.08 | 2.80 | 1.83 |

Table 5: Ablation study of module design. The *Cost* shows the token required to complete the ScanRefer-Justice dataset (Million). **Bold** denotes best performance.

| ID | Refer-Judge | | | Agreement ↑ | Precision ↑ | Recall ↑ | F1 ↑ | RMSE ↓ | MAE ↓ | Cost ↓ |
| | Log. | Obs. | Ref. | | | | | | | |
|---|---|---|---|---|---|---|---|---|---|---|
| 1 | | ✓ | ✓ | 81.55 | **83.44** | 82.18 | 82.80 | 2.91 | 1.93 | 108.78 |
| 2 | ✓ | ✓ | | 81.72 | 83.15 | 83.00 | 83.08 | 2.80 | 1.83 | 82.41 |
| 3 | ✓ | | | 54.20 | 58.93 | 50.50 | 54.39 | 5.10 | 3.72 | **9.41** |
| 4 | ✓ | ✓ | ✓ | **82.77** | 82.95 | **85.77** | **84.33** | **2.69** | **1.71** | 112.79 |

methods benefit significantly from the purified training data, achieving consistent improvements. For example, 3DVLP+Refer-Judge outperforms the baseline by +1.6% in Overall Acc@0.5, achieving competitive 3DVG performance.

To better understand the effects of toxic data, we further separate the validation set into purified and toxic subsets using Refer-Judge. As reported in Table 2b, the performance gain becomes even more evident, with +3.10%, +1.84% and +1.21% improvements in Overall Acc@0.5, respectively. On the other hand, the baseline model achieves better performance on toxic samples, which are inherently unable to accurately predict. This discrepancy may arise because the original training set provides implicit priors aligned with the validation distribution, particularly benefiting the toxic samples, exhibiting inflated scores, which highlight the critical role of toxic data exposing in making reliable evaluations for 3DVG. More results and analysis are left in Appendix C.

## 4.3 Ablation study

**Ablation of the Jury System.** We first examine the effectiveness of the Jury system in Refer-Judge. As shown in Table 3, the best performance is achieved when all four jurors jointly assess the sample from multiple perspectives, including linguistic logic and visual observation. This confirms that structured decomposition and distributed evaluation significantly enhance targeted reasoning over complex scene-level 3DVG data.

**Ablation of the Judge System.** Next, we analyze the contribution of our proposed Corroborative Refinement strategy (CR) within the Judge module. As shown in Table 4, our method outperforms alternative strategies such as Self-consistency with Chain-of-Thought (CoT-SC) [34], ranking and re-evaluation of problematic results (GoT [5]), and naive score averaging. While CoT-SC yields a comparable agreement score, its reliance on majority voting often leads to polarized decisions, which fails on fine-grained scoring metrics like RMSE and MAE. These findings highlight the importance of adaptive refinement through auxiliary information in understanding the complex 3D scene.

**Ablation of Model Design.** Finally, we summarize the impact of each component on performance and cost. As presented in Table 5, visual observation (e.g., Distinguishability and Ambiguity jurors) contributes the most to performance but also dominates token cost due to scene-level image analysis. In contrast, logic-related assessments have a relatively lower impact, as visual jurors may implicitly assess textual logic. Meanwhile, the refinement in the Judge system provides consistent improvements, helping the Refer-Judge achieve human-comparable performance in uncovering paradoxical and ambiguous 3DVG data. More detailed results are left in Appendix C.

## 5 Conclusion

In this paper, we introduce Refer-Judge, a novel framework for uncovering paradoxes and ambiguities in 3DVG datasets, along with a high-quality benchmark for evaluating data-centric methods. By leveraging a Jury-and-Judge Chain-of-Thought paradigm, Refer-Judge effectively harnesses the

reasoning and perceptual capabilities of MLLMs, achieving human-comparable performance in identifying toxic samples. Moreover, baseline models trained on the purified data consistently exhibit improved performance, underscoring the practical value of exposing and filtering flawed annotations. We hope this work offers a new lens on dataset reliability in 3D scene understanding and serves as a foundation for more trustworthy training and evaluation practices.

**Limitations.** While Refer-Judge demonstrates strong effectiveness, its computational cost remains a barrier to large-scale deployment. Future work could focus on optimizing the judgment process and leveraging more efficient MLLM architectures to enhance scalability and extend applicability to larger and more diverse 3D datasets.

**Acknowledgment.** This work is supported by the "Pioneer" and "Leading Goose" R&D Program of Zhejiang (2024C01020, 2025C01091), Key Technology Breakthrough Plan Project of "Science and Technology Innovation Yongjiang 2035" (2025Z058, 2024Z295).

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

# A  ScanRefer-Justice Dataset

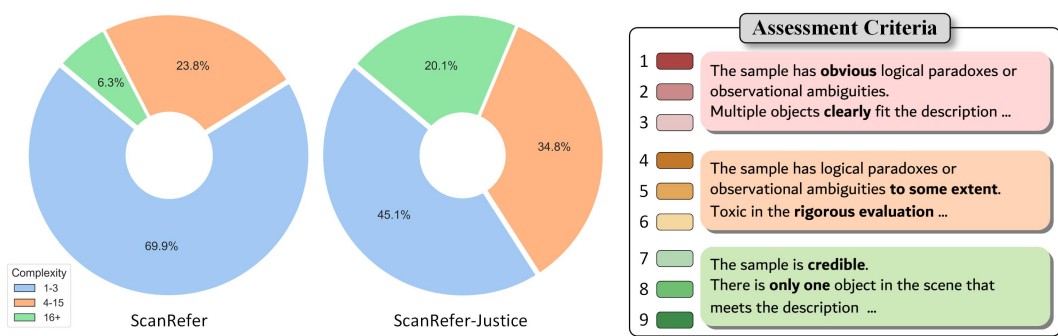

Figure 4: Complexity distribution of ScanRefer and ScanRefer-Justice datasets.

Figure 5: Assessment Criteria for annotating ScanRefer-Justice.

To evaluate the effectiveness of our proposed method in identifying toxic samples, we introduce a new benchmark dataset: **ScanRefer-Justice**. This dataset is constructed upon the widely used *ScanRefer* dataset [7], which contains 51,583 referring expressions for objects from ScanNet [10] scenes.

## A.1  Data Selection

Due to the cost and complexity of conducting rigorous human evaluation across the entire ScanRefer dataset, it is impractical to manually verify all samples while ensuring annotation quality. Moreover, we observe that the occurrence of toxic data is highly correlated with scene complexity. Over-representation of simple samples (i.e., only a few objects of the same class) can lead to skewed data distribution and biased algorithm evaluation.

To better reflect this complexity, we categorize the ScanRefer samples into three levels based on the number of same-class objects with the target one: *Normal* (1–3 instances), *Hard* (4–15), and *Complex* (16 or more). As shown in Figure 4, we choose 3,001 data from the ScanRefer training and validation sets to form **ScanRefer-Justice**, covering 162 scenes. The sampling focuses on increasing coverage of more complex scenes to better assess the performance of 3DVG data judgment algorithms.

Specifically, ScanRefer-Justice significantly rebalances the data composition compared to ScanRefer. While *complex* samples constitute only 6.3% in ScanRefer, they account for over 20% in ScanRefer-Justice. Combined with *hard* samples, more than 50% of the dataset consists of challenging examples, capturing a wider range of possible sources of annotation errors.

## A.2  Data Collection

To ensure consistency and rigor in data labeling, we employ trained internal annotators (graduate students), rather than outsourcing, as was done in the original ScanRefer annotation process. The annotation pipeline consists of two stages: (i) Annotation and (ii) Verification.

**Annotation.** Annotators first observe the complete 3D point cloud scene using Open3D visualization tools. This step excludes both the referring expression and ScanNet's object labels to reduce prior bias. Observers are required to spend at least 20 seconds examining the scene to ensure a comprehensive understanding.

Next, expanding the annotation process of ScanRefer [7], we provide up to six 2D reference views per sample to compensate for incomplete details in the 3D point cloud reconstructions, where three images containing the target object and three with distractor objects of the same category. Category definitions are aligned with the original setting of ScanRefer. Meanwhile, the corresponding referring expression is presented as on-screen subtitles.

Based on this information, annotators assess each sample's quality from two perspectives: the Logical Completeness of the description and the Uniqueness of the visual context. Throughout the process, the annotator can revisit the 3D scene and consult additional ScanNet frames for cross-referencing.

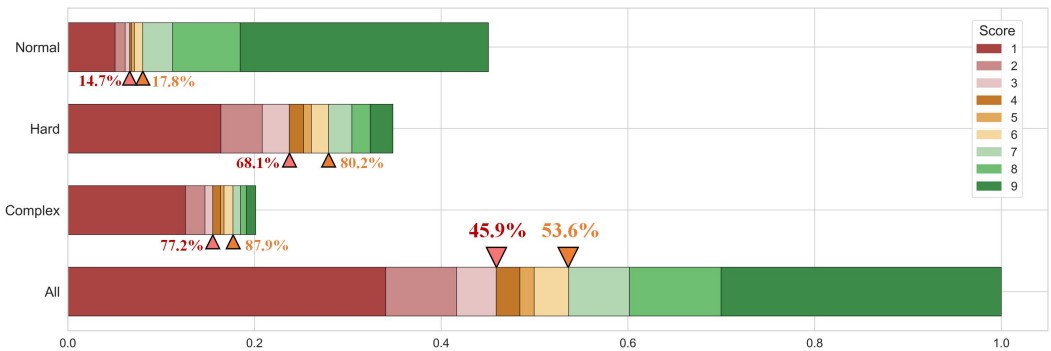

Figure 6: Distribution of reliability scores in the ScanRefer-Justice dataset.

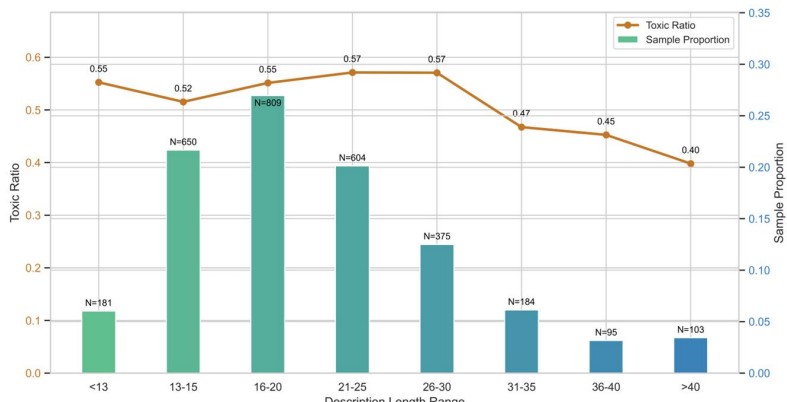

Figure 7: Relationship between reliability score and sentence length in ScanRefer-Justice dataset.

To ensure consistency in scoring, we require annotators to follow a 9-point Likert scale based on predefined *Assessment Criteria* (higher scores mean higher data quality), as shown in Figure 5.

**Verification.** After the initial round, the annotators carry out cross-checks to validate the consistency of the annotations and estimate human error rates in toxic data identification (as reported in Table 1). During this stage, the previous annotation score is displayed together with the referring expression.

### A.3 Data Statistics

Figure 6 illustrates the score distribution in ScanRefer-Justice. Benefits to our targeted sampling, the dataset achieves a balanced distribution of reliable and toxic samples. Specifically, 45.9% of samples are rated as clearly erroneous, 7.7% as moderately flawed, and 46.4% as clearly reliable. Most scores cluster around 1 or 9, indicating strong confidence among annotators. Samples rated in the ambiguous 4–6 range are notably sparse. Additionally, it is clear that toxic data appears more frequently in scenes with higher complexity.

In addition, we examine the relationship between referring expression length and the quality of 3DVG data, as shown in Figure 7. Interestingly, short expressions (<15 tokens) and medium-length ones (16–30 tokens) exhibit similar error rates, likely due to compensating effects between scene simplicity and annotation carelessness. Logically, descriptions with longer than 30 tokens show a marked improvement in reliability, suggesting that detailed language helps disambiguate targets in 3DVG. Nevertheless, longer descriptions may impose a cognitive burden on annotators. Leveraging LLMs to provide adaptive, guided assistance during annotation may be a promising direction for future 3DVG dataset construction.

Table 6: Quantitative results on ScanRefer test set.

| Method | Unique ↑ | | Multiple ↑ | | Overall ↑ | |
|---|---|---|---|---|---|---|
| | Acc@0.25 | Acc@0.5 | Acc@0.25 | Acc@0.5 | Acc@0.25 | Acc@0.5 |
| InstanceRefer [44] | 77.82 | 66.69 | 34.57 | 26.88 | 44.27 | 35.80 |
| 3DVG-Transformer [51] | 75.76 | 55.15 | 42.24 | 29.33 | 49.76 | 35.12 |
| D3Net [8] | 79.23 | 68.43 | 39.05 | 30.74 | 48.06 | 39.19 |
| D-LISA [46] | 81.95 | 69.00 | 49.75 | 39.67 | 56.97 | 46.25 |
| ScanRefer [7] | 68.59 | 43.53 | 34.88 | 20.97 | 42.44 | 26.03 |
| + *Refer-Judge* | 73.45(+4.86) | 47.16(+3.63) | 35.36(+0.48) | 21.25(+0.28) | 43.90(+1.46) | 27.06(+1.03) |
| 3DVLP [47] | 78.24 | 62.98 | 45.32 | 34.05 | 52.70 | 40.54 |
| + *Refer-Judge* | 79.39(+1.15) | 65.46(+2.48) | 46.51(+1.19) | 34.91(+0.86) | 53.88(+1.18) | 41.76(+1.22) |

## B Details of Metric Calculations

To evaluate the ability of the proposed method to identify toxic samples in 3DVG datasets, we first formulate the task as a binary classification problem. Notably, since we are more concerned with the identification of toxic samples, toxic samples are considered as the **positive class**, and valid samples as the **negative class**. We use TP, FP, TN, and FN to represent the true positive, false positive, true negative, and false negative, respectively.

We define toxic samples as those with human-annotated scores between 1-6, including both *obviously untrustworthy* (score between 1-3) and *partially flawed* samples (score between 4-6). Valid samples are those with scores in the range of 7–9. The calculation of agreement, precision, recall, and F1-score are as follows:

$$
\begin{aligned}
\text{Agreement} &= (TP + TN) / (TP + FP + TN + FN) \\
\text{Precision} &= TP / (TP + FP) \\
\text{Recall} &= TP / (TP + FN) \\
\text{F1-score} &= (2 \cdot TP) / (2 \cdot TP + FP + FN)
\end{aligned}
\tag{8}
$$

To further evaluate the alignment between our method and human assessments on a finer scale, we additionally treat this task as a regression problem and report the following metrics:

$$
\text{RMSE} = \sqrt{\frac{1}{N} \sum_{i=1}^{N} (s_i - \hat{s}_i)^2}, \quad \text{MAE} = \frac{1}{N} \sum_{i=1}^{N} |s_i - \hat{s}_i|
\tag{9}
$$

where $s_i$ is the human-annotated score and $\hat{s}_i$ is the predicted score for the $i$-th sample.

## C Additional Experiments

**Comparison on the ScanRefer Test Benchmark.** To further validate the effectiveness of filtering toxic data, we report method performance on the ScanRefer test set. As shown in Table 6, after training on purified data organized via Refer-Judge, both baseline models exhibit consistent performance gains, with +1.03% and +1.22% in Overall Acc@0.5, confirming the stability of our method. Notably, combined with the validation results in Table 2, the performance improvements are more pronounced in scenes where only one object of the target category exists *Unique*, compared to *Multiple* samples (with multiple similar objects). This is likely because the original training data contains distributional biases that help models overfit toxic samples. Correspondingly, as shown in Figure 6, toxic samples are more concentrated in complex scenes, allowing baseline models to gain an extra advantage in Multiple-type evaluations.

**Comparison on the Purified Validation Set.** To better analyze the impact of toxic data on algorithm evaluation, we extend the analysis in Table 2 by progressively decreasing the threshold in toxicity judgment, thereby including a broader range of potentially toxic samples. As presented in Table 7, the proportion of identified toxic data rises from 7.6% to 40.6% when the Refer-Judge threshold increases

Table 7: Quantitative results on toxic and purified data under different thresholds. The '**Thr.**' indicates the score threshold and the resulting proportion of toxic data under that threshold.

| Method | Thr. | Toxic data ↓ | | Unique (purified) ↑ | | Multiple (purified) ↑ | | Overall (purified) ↑ | |
|---|---|---|---|---|---|---|---|---|---|
| | | Acc@0.25 | Acc@0.5 | Acc@0.25 | Acc@0.5 | Acc@0.25 | Acc@0.5 | Acc@0.25 | Acc@0.5 |
| ScanRefer | | 20.44 | 13.12 | 76.91 | 50.57 | 34.77 | 21.78 | 43.60 | 27.81 |
| + *Refer-Judge* | 1 | 17.96(-2.48) | 12.84(-0.28) | 79.52(+2.61) | 55.40(+4.83) | 35.73(+0.96) | 24.27(+2.49) | 44.91(+1.31) | 30.79(+2.98) |
| 3DVLP | ~7.6% | 22.69 | 17.36 | 84.65 | 68.27 | 44.58 | 34.38 | 52.97 | 42.17 |
| + *Refer-Judge* | | 22.41(-0.28) | 14.43(-2.93) | 86.70(+2.05) | 70.42(+2.15) | 46.49(+1.91) | 36.01(+1.63) | 54.91(+1.94) | 43.22(+1.05) |
| ScanRefer | | 21.69 | 14.58 | 76.89 | 50.57 | 34.96 | 21.8 | 43.91 | 27.94 |
| + *Refer-Judge* | ≤ 2 | 18.76(-2.93) | 13.67(-0.91) | 79.50(+2.61) | 55.41(+4.84) | 36.07(+1.11) | 24.44(+2.64) | 45.33(+1.42) | 31.04(+3.10) |
| 3DVLP | ~9.3% | 22.91 | 17.75 | 85.97 | 70.02 | 46.44 | 36.02 | 54.86 | 43.01 |
| + *Refer-Judge* | | 22.11(-0.80) | 15.46(-2.29) | 86.41(+0.44) | 72.27(+2.25) | 47.14(+0.70) | 37.43(+1.41) | 55.50(+0.64) | 44.85(+1.84) |
| ScanRefer | | 22.58 | 16.03 | 77.02 | 50.66 | 37.34 | 22.88 | 46.99 | 29.63 |
| + *Refer-Judge* | ≤ 3 | 21.53(-1.05) | 15.60(-0.43) | 79.61(+2.59) | 55.59(+4.93) | 38.59(+1.25) | 25.91(+3.03) | 48.56(+1.57) | 33.12(+3.49) |
| 3DVLP | ~21.1% | 27.57 | 21.00 | 86.18 | 70.53 | 49.68 | 38.48 | 58.55 | 46.27 |
| + *Refer-Judge* | | 25.85(-1.72) | 18.83(-2.17) | 86.52(+0.34) | 72.42(+1.89) | 50.84(+1.16) | 40.63(+2.15) | 59.51(+0.96) | 48.36(+2.09) |
| ScanRefer | | 24.23 | 17.00 | 77.12 | 50.71 | 38.97 | 23.39 | 49.41 | 30.87 |
| + *Refer-Judge* | ≤ 4 | 23.46(-0.77) | 17.28(+0.28) | 79.65(+2.53) | 55.72(+5.01) | 40.51(+1.54) | 26.73(+3.34) | 51.22(+1.81) | 34.66(+3.79) |
| 3DVLP | ~40.6% | 29.32 | 22.91 | 86.3 | 70.71 | 52.34 | 40.42 | 61.63 | 48.70 |
| + *Refer-Judge* | | 28.36(-0.96) | 21.85(-1.06) | 86.64(+0.34) | 72.55(+1.84) | 53.77(+1.43) | 42.69(+2.27) | 62.76(+1.13) | 50.86(+2.16) |

Table 8: Comprehensive ablation study of module design. The *Cost* shows the token required to complete the ScanRefer-Justice dataset (Million). **Bold** denotes best performance.

| ID | Refer-Judge | | | | | | Agreement ↑ | Precision ↑ | Recall ↑ | F1 ↑ | RMSE ↓ | MAE ↓ | Cost ↓ |
|---|---|---|---|---|---|---|---|---|---|---|---|---|---|
| | $J_l$ | $J_c$ | $J_d$ | $J_a$ | $J_d^{mag}$ | $J_a^{mag}$ | | | | | | | |
| 1 | | | | | | | **82.77** | 82.95 | **85.77** | **84.33** | **2.69** | **1.71** | 112.79 |
| 2 | ✗ | | | | | | 82.53 | 83.87 | 83.81 | 83.84 | 2.73 | 1.77 | 111.36 |
| 3 | | ✗ | | | | | 81.75 | 82.92 | 83.44 | 83.18 | 2.88 | 1.87 | 110.21 |
| 4 | | | ✗ | | ✗ | | 80.86 | **85.28** | 78.09 | 81.53 | 2.97 | 1.89 | 50.83 |
| 5 | | | | ✗ | | ✗ | 62.41 | 64.00 | 69.65 | 66.71 | 3.74 | 2.68 | 78.78 |
| 6 | | | | | ✗ | | 81.61 | 83.33 | 82.49 | 82.91 | 2.82 | 1.81 | 85.13 |
| 7 | | | | | | ✗ | 82.67 | 83.44 | 84.76 | 84.01 | 2.73 | 1.74 | 110.07 |
| 8 | ✗ | ✗ | | | | | 81.55 | 83.44 | 82.18 | 82.80 | 2.91 | 1.93 | 108.78 |
| 9 | | | | | ✗ | ✗ | 81.72 | 83.15 | 83.00 | 83.08 | 2.80 | 1.83 | 82.41 |
| 9 | | | ✗ | ✗ | ✗ | ✗ | 54.20 | 58.93 | 50.50 | 54.39 | 5.10 | 3.72 | **9.41** |

from 1 to 4. While a lenient threshold inevitably misjudges more valid samples, we ensure all methods are evaluated on the same purified validation set to keep fairness. The results demonstrate that across all toxicity thresholds, models trained with Refer-Judge consistently outperform the corresponding baselines, confirming the robustness of our method. Interestingly, we observe non-uniform margins between purified and baseline results. For instance, at threshold 1 (strict), 3DVLP+Refer-Judge outperforms the baseline by +1.05% in Overall Acc@50, whereas the gain increases to +2.16% at threshold ≤ 4 (lenient). This highlights that beyond the identification of toxic data, establishing a standardized evaluation framework is essential for enabling robust and comprehensive method assessment.

**Comprehensive Ablation of Module Design.** We then expand the model design shown in Table 5 to investigate the impact of each module in the Refer-Judge framework. Experiments 2–5 in Table 8 examine the removal of each individual juror within the Jury system. The results show a consistent performance drop, confirming all four perspectives contribute indispensably to toxic data identification. Among them, the Ambiguity Juror proves most impactful, highlighting a prevalent issue in 3DVG annotation: the failure to verify the presence of similar misleading objects in the scene. Meanwhile, comparing experiments 1 and 6, the introduction of Corroborative Refinement in $J_d^{mag}$ improves the Agreement score by +1.16%, demonstrating the benefit of using auxiliary information to guide and correct flawed judgments arising from incomplete observation.

# D Case Study

Finally, we present some representative cases to intuitively illustrate the decision-making process of Refer-Judge on 3DVG samples. On the one hand, we show some cases from the ScanRefer dataset [7], which respectively correspond to *Normal*, *Hard*, and *Complex* scenarios.

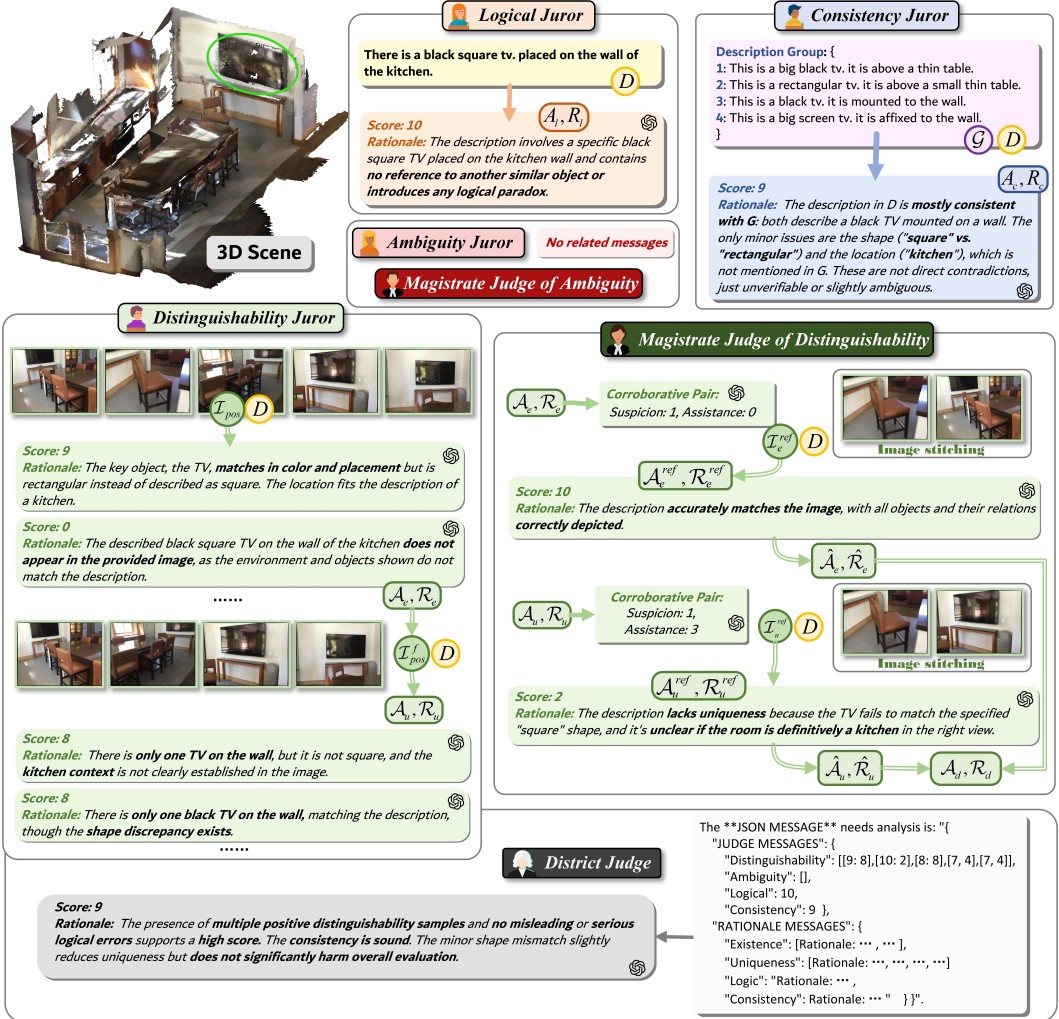

Figure 8: A *normal* case where only one TV is involved in the scene.

**Detailed Analysis.** Although this is a relatively simple case, several interesting details emerge. For instance, when analyzing the second view, $J_d$ fails to detect a matching object because the TV is only partially visible at the edge of the frame. This viewpoint-induced failure is subsequently corrected during the refinement process provided by $J_d^{mag}$. Moreover, we also observe errors caused by hallucination. For example, in its second step, $J_d^{mag}$ incorrectly rejects the TV in the image for not satisfying the specified "square" shape.

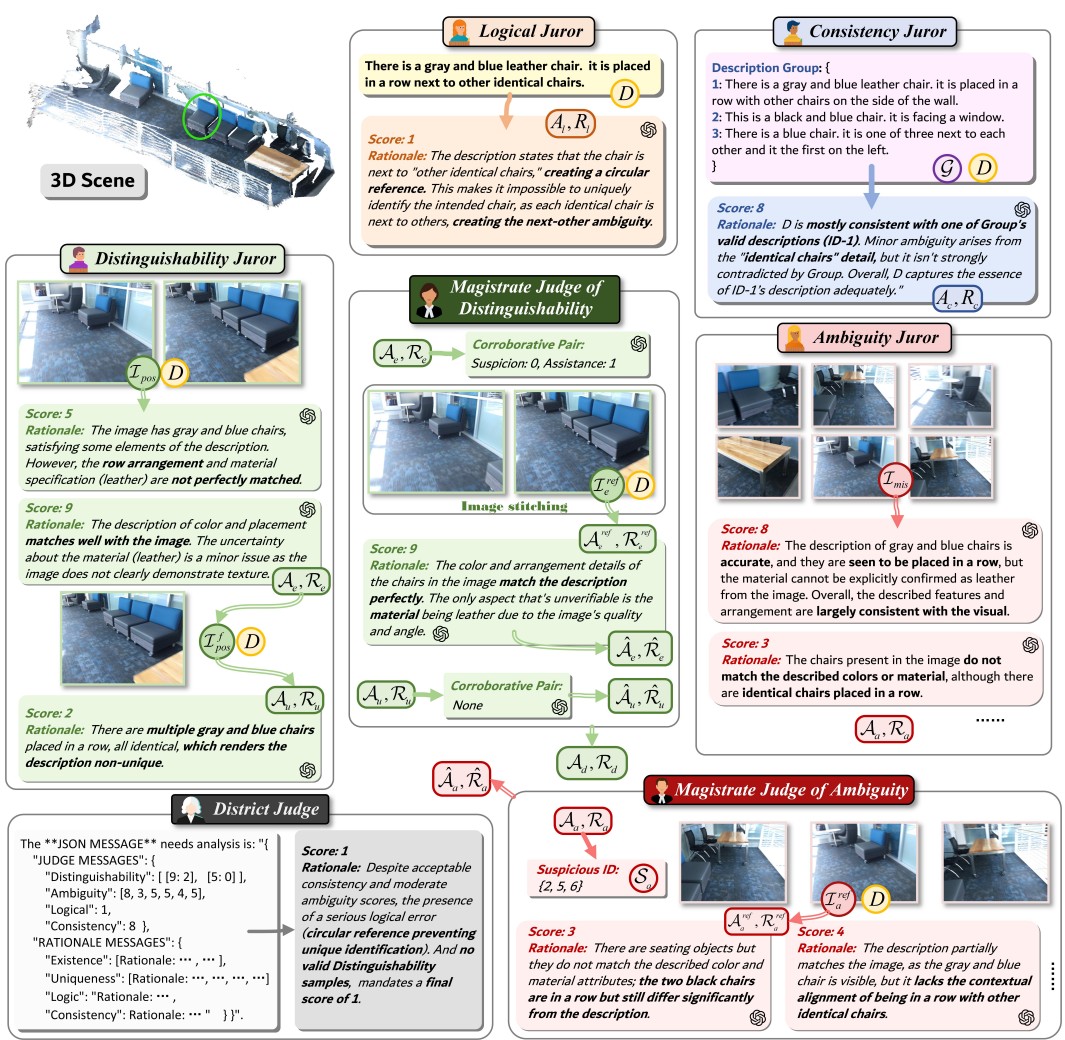

Figure 9: A *hard* case where Refer-Judge provides correct assessment in both paradox and ambiguity identification.

**Detailed Analysis.** As a case exhibiting both paradox and ambiguity, multiple components contribute to the final judgment. For instance, $J_l$ provides a direct evaluation of the textual paradox, while $J_d$ assesses the object's uniqueness. This demonstrates the advantage of Refer-Judge in analyzing problematic samples from multiple complementary perspectives.

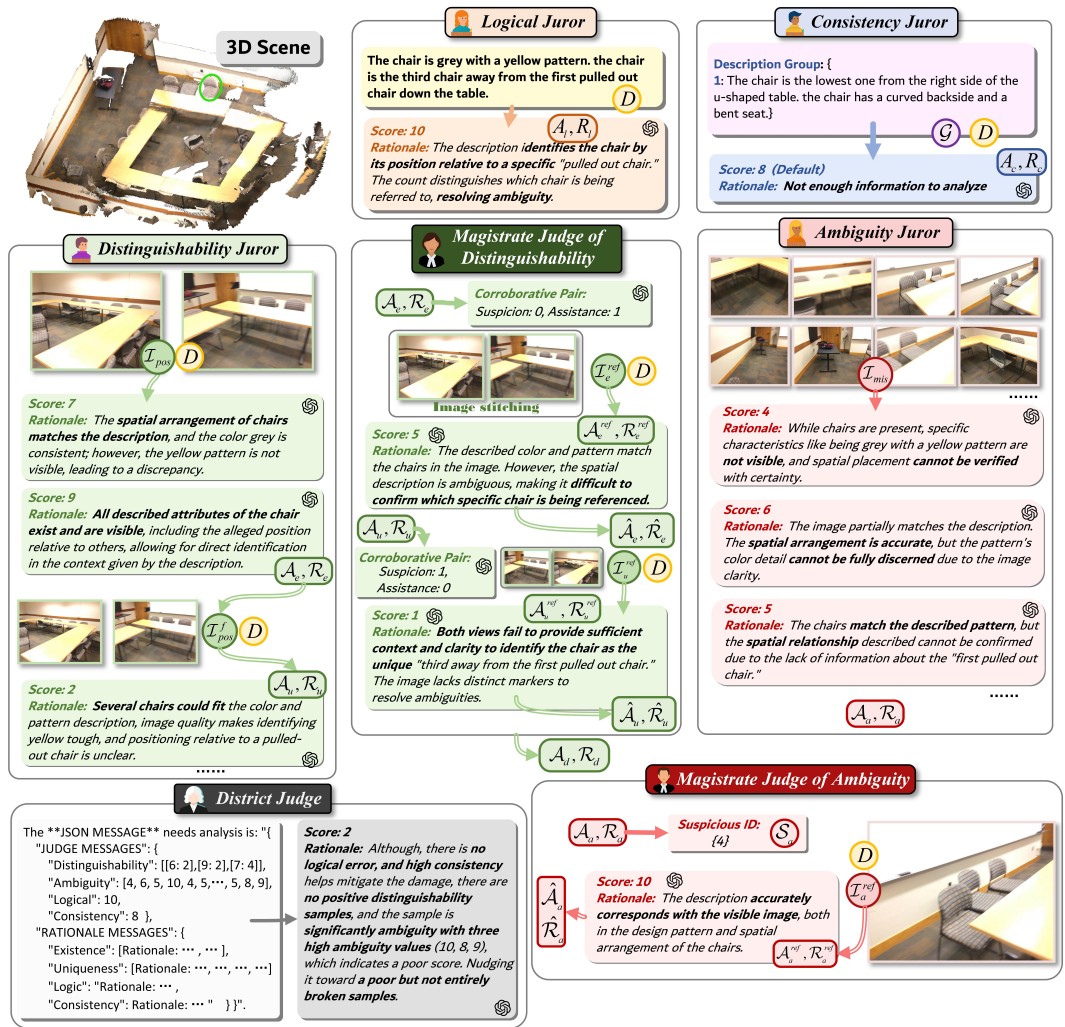

Figure 10: A *complex* case where over 20 similar objects introduce significant visual ambiguity.

**Detailed Analysis.** In this more complex scene, $J_a$ processes a larger batch of images and thus plays a more significant role in the overall decision-making process. However, the analyses from $J_a$ and $J_a^{mag}$ also reveal that even for two highly similar images ($I_m^3$ and $I_m^4$), their corresponding evaluations may differ substantially. This further highlights the importance of maintaining evaluation consistency within such an agentic system.

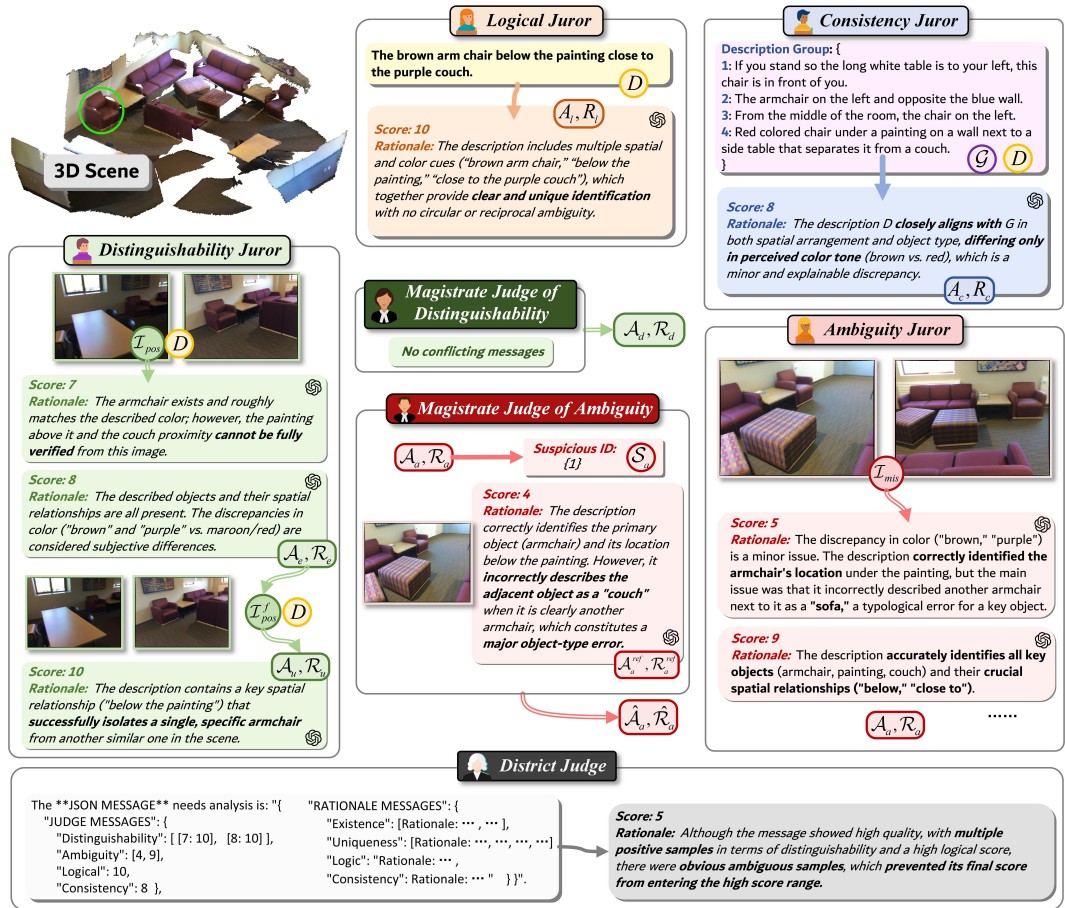

Figure 11: A case passes the inspection of Nr3D but is actually ambiguous.

On the other hand, we further extend Refer-Judge to the Nr3D dataset [2], which additionally provides examiner (listener) annotations assessing the quality of the original labels. Despite this stricter verification setting, Refer-Judge still uncovers toxic data within Nr3D, including cases where samples consistently judged as correct by annotators are actually problematic, and instances flagged as erroneous by examiners are in fact solid. These extended case studies further demonstrate the effectiveness and generalizability of our proposed framework.

**Detailed Analysis.** Although this is a relatively simple scene, the evident ambiguity was not detected by annotators, revealing the widespread presence of toxic data in manually labeled 3DVG datasets. Notably, during the ambiguity analysis, $I_m^1$ captures only a partial view of the painting, leading to a low score from $J_a$, which $J_a^{mag}$ also fails to correct. This observation underscores the current limitations of our system.

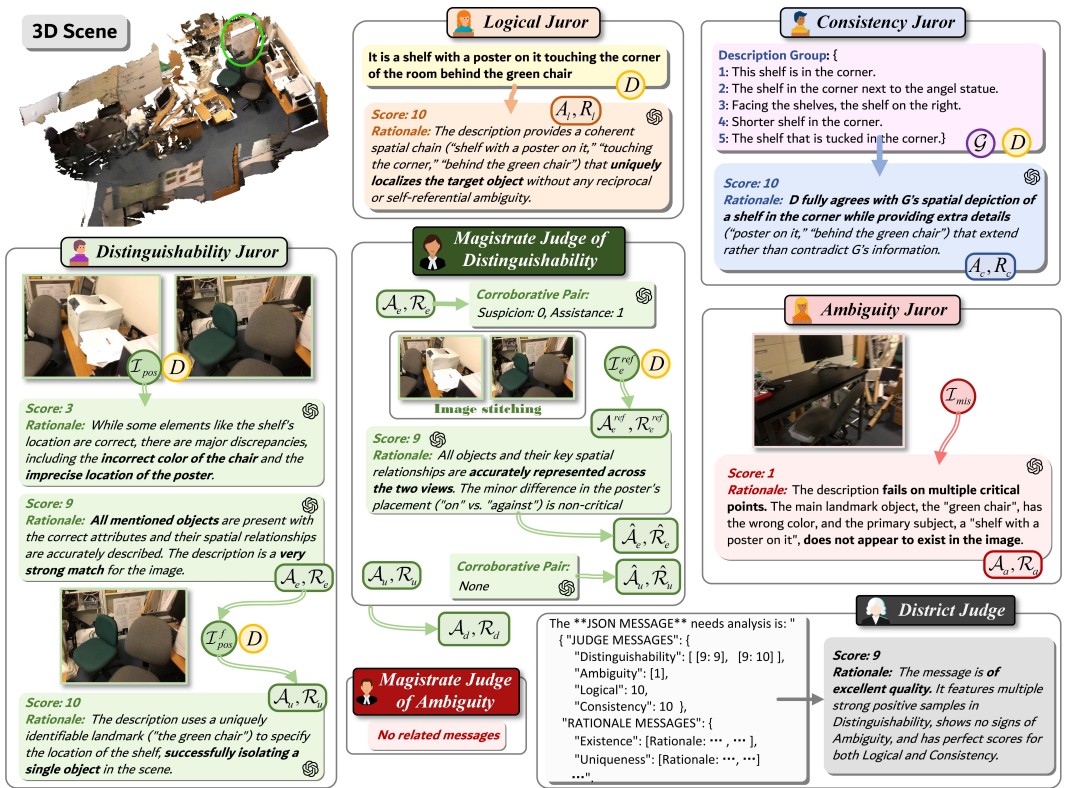

Figure 12: A case marked as failed in Nr3D (due to speaker-listener mismatch) but turned out to be valid on re-check.

**Detailed Analysis.** This is an interesting case where the examiner assigned a low-confidence mark, even though the referring expression was already quite detailed. Through our Refer-Judge framework, we observe that the high score given by $J_d$ to $I_p^2$ leads to a final positive evaluation for this sample.

