# OpenReview forum: "Jury-and-Judge Chain-of-Thought for Uncovering Toxic Data in 3D Visual Grounding"
_NeurIPS.cc/2025/Conference — NeurIPS 2025 poster_

### Official Review · Reviewer_KTYR · 2025-07-02

**Clarity:** 1
**Significance:** 2
**Originality:** 2
**Rating:** 2
**Confidence:** 4

**Summary:**

This paper proposes Refer-Judge, a reasoning-based framework for improving data quality in 3D visual grounding tasks. The core idea is to apply a two-stage Jury-and-Judge Chain-of-Thought (CoT) mechanism using Multimodal Large Language Models (MLLMs) to detect and refine “toxic” (logically inconsistent or ambiguous) data. Jurors provide diverse judgments, which are consolidated by a Judge module using a "Corroborative Refinement" step. The refined dataset is then shown to improve model performance across various evaluation splits of the ScanRefer dataset.

**Questions:**

See Weaknesses

**6. Computation and Cost Concerns**
Running MLLMs over large-scale 3D datasets (like ScanRefer) can be computationally expensive. How many total model calls are required?
Have the authors estimated the cost in terms of GPU-hours or dollar cost for filtering ScanRefer?
Would it be more efficient or cost-effective to rely on human annotators instead? At the very least, a comparison of cost and scalability between human and MLLM-based evaluation would add realism to the proposed solution.

**Ethical Concerns:**

["NO or VERY MINOR ethics concerns only"]

**Limitations:**

Yes, however, it is a significant limitation, specifically with marginal gain.

**Paper Formatting Concerns:**

No formatting concerns.

**Quality:**

2

**Strengths And Weaknesses:**

---
**Strengths**

1- Targeted Problem: The paper addresses a real and often overlooked challenge in 3D visual grounding—noisy and ambiguous annotations that arise from scene complexity and limited field of view.

2- High-Level Conceptual Framing: The use of a jury-and-judge metaphor combined with Chain-of-Thought reasoning is intuitive and aligns well with recent trends in using LLMs for structured evaluation.

3- Empirical Signals: The purified dataset shows measurable improvements across multiple performance metrics and split types (unique, multiple, and toxic data), suggesting some practical benefit.

---
**Weaknesses and Suggestions for Improvement**

**1. Limited Methodological Novelty:**
The proposed pipeline essentially repurposes existing MLLMs for multi-perspective reasoning. While practically useful, the approach lacks significant algorithmic novelty. It primarily orchestrates structured prompting with strong pre-trained models.

The reliance on MLLMs without any architectural or methodological contribution makes it unclear how much of the performance gain comes from the framework itself versus the capabilities of the underlying model.


**2. Overclaims and Writing Style:**
The writing often overstates the technical depth (e.g., referring to structured prompting as a “framework” with “corroborative refinement”) and lacks precision in describing concrete steps.
The abstract and introduction are heavy on conceptual metaphors but thin on details, which hinders readability and makes the contributions harder to parse.


**3. Missing or Incomplete Related Work**
The paper omits several key and recent works in 3D visual grounding [1,2,3]
These works provide strong baselines and theoretical frameworks that should have been acknowledged and compared against.


**4. Experimental Limitations**
Single-Dataset Evaluation: The framework is only evaluated on ScanRefer, while Nr3D and Sr3D are more controlled benchmarks for reasoning and ambiguity. Particularly, Nr3D’s viewpoint splits and distractor-based setups align well with the paper’s motivation and should not be omitted.


Table 1 Clarity:
The input modality is unclear—are models using raw point clouds, RGB images, or both?
How are Precision, Recall, and RMSE computed? What is the ground truth (e.g., bounding box centers or full masks)?
The table appears to mix classification-style and regression-style metrics without a proper explanation.

Table 2a:
Only two baselines are fine-tuned using purified data. Given the broad landscape of 3DVG models, this is insufficient. The omission of ConcreteNet, a strong recent model, is especially concerning.

Table 2b:
The results on toxic data are confusing. For instance, purified models sometimes perform worse than their baseline counterparts. Why is that the case?
A key point is missing: if the evaluation set is toxic, then models trained on purified data may appear worse due to mismatched assumptions. A random baseline or distractor-aware baseline could help contextualize the drop.

**5. Ambiguity in the Dataset Filtration Process**
The paper mentions "positive sets" (e.g., Line 172) without clarifying how they are constructed. Are they human-verified, heuristically generated, or filtered by some confidence threshold?
The filtration process should be clearly specified: how many MLLM calls per sample? Are jurors prompted independently or jointly?
If MLLMs are trusted enough to curate data, why not simply rely on them during inference as well? This tradeoff deserves discussion.

[1] 3DRP-Net: 3D Relative Position-aware Network for 3D Visual Grounding
[2] Viewpoint-Aware Visual Grounding in 3D Scenes
[3] CoT3DRef: Chain-of-Thoughts Data-Efficient 3D Visual Grounding

---

> ### Author Rebuttal · Authors · 2025-07-31
>
> We sincerely thank the reviewer for the detailed feedback and the opportunity to clarify and strengthen our submission. Below, we address each of the raised concerns.
>
> **Q1: Limited Methodological Novelty**
>
> We respectfully disagree that the proposed method merely renames standard reasoning mechanisms. While Refer-Judge is rooted in CoT-based LLM reasoning, it is, to the best of our knowledge, the first agentic, deliberation-based system aimed specifically at evaluating complex dataset quality in 3DVG. We believe this Jury-and-Judge CoT represents a structured and extensible paradigm for LLM-as-a-Judge applications. Its value lies in a novel and structured formulation of LLM-based data auditing—a dimension increasingly relevant in the era of foundation models.
>
> We appreciate prior efforts such as GoT, which explore topological variations of CoT graphs. However, as we stated in our related works, *most CoT-based methods are tailored for narrow-scope tasks, including arithmetic reasoning (like GoT) or short image-text QA (like CoCoT, DCoT)*. These methods demonstrate the existence of complex CoT topologies, but not their utility in executing open-ended, multi-perspective tasks like data auditing in 3DVG.
>
> Our framework introduces:
>
> - Modular Role-Driven Reasoning: Jurors are semantically differentiated agents, each targeting a key perspective. This design goes beyond ensemble prompting, aligning with how real-world juries solicit diverse perspectives. Unlike GoT-style parallel agents, Refer-Judge enforces cross-perspective reasoning and inter-agent role separation.
> - Corroborative Refinement: The judge layer is not a majority vote but a structured aggregation then re-evaluation mechanism, including conflicts recognition, information reorganizing, and corroborative scoring. This mitigates the effects of unreliable observations. As shown in Table 4, our refinement significantly outperforms both GoT and CoT-SC, underscoring its effectiveness.
>
> While we acknowledge that Refer-Judge is not focused on in-depth exploration of CoT topology, we believe this method represents methodological innovations in LLM-as-a-Judge field.
>
> Meanwhile, to verify the role of each component in Refer-Judge, we also conducted extensive ablation experiments. As shown in Table 8, the contribution of each module to toxic data uncovering is clearly distinguishable.
>
> **Q2: Overclaims and Writing Style**
>
> We appreciate this point and will revise the paper to: (1) reduce metaphorical overstatements, and (2) clarify all modules with more technical precision.
>
> That said, we would like to clarify our intent behind the “Jury-and-Judge” analogy. We believe this is not a superficial metaphor, but a natural and useful abstraction for the emerging MLLM-as-a-Judge field. It captures the essence of multi-perspective deliberation and structured refinement, echoing real-world adjudication processes that aggregate diverse evidence and viewpoints.
>
> Moreover, as detailed in our response to Q1, the Corroborative Refinement process is not merely “structured prompting,” but a new refinement method that performs inter-judgment consistency checking.
>
> **Q3: Missing or Incomplete Related Work**
>
> Thank you for highlighting this. These are indeed important and highly relevant theoretical precedents. We will revise the Related Work section to explicitly incorporate and discuss the following recent advances:
>
> - 3DRP-Net: Enhances 3DVG by modeling relative spatial relations
> - VPP-Net: Emphasizes viewpoint positioning
> - CoT3DRef: Introduces CoT into 3DVG with a Seq2Seq approach
>
> **Q4: Experimental Limitations**
>
> **Part 1: Dataset Scope:**
>
> We acknowledge that evaluating solely on ScanRefer may limit the perceived generalizability of Refer-Judge. Therefore, due to time constraints, we first conducted a small-scale study on the Nr3D dataset. Manual review by human experts confirmed that toxic samples also exist.
>
> Encouragingly, Refer-Judge aligned well with expert judgment in these cases. For example, for sample #23273 (*The brown armchair below the painting close to the purple couch*), although the original annotation marked the listener’s pick as correct, the scene (scene0520_00) actually contains two valid armchairs below different paintings, making the expression ambiguous. Refer-Judge successfully identified this.
>
> Interestingly, we also observed cases where annotations were marked as failed in Nr3D (due to speaker-listener mismatch) but turned out to be valid on re-check. For instance, sample #39432 (*It is a shelf with a poster on it touching the corner of the room behind the green chair*) can be clearly grounded to a unique shelf (seen in 000500.jpg of scene0565_00) by both experts and Refer-Judge.
>
> We will include these findings in the revised version, and we welcome future work extending Refer-Judge to other datasets.
>
> **Part 2: Clarification on Table 1**
>
> We thank the reviewer for requesting clarification. As described in Section 3.1, Refer-Judge uses RGB images as the primary observation modality throughout the reasoning process.
>
> Regarding the evaluation metrics, we provide a detailed definition in Appendix B. Specifically:
> - For classification-style metrics, we adopt human expert annotations as binary ground truth: we define toxic samples as those scoring 1–6, while valid samples are those scoring 7–9. This binary formulation enables us to assess whether Refer-Judge can reliably match human-level identification of flawed annotations.
> - For regression-style metrics, since Refer-Judge produces scores in the same 1–9 range as the ScanRefer-Justice expert annotations, RMSE offers a fine-grained view of numerical alignment.
>
> **Part 3: Table 2a Coverage**
>
> We agree with the reviewer that demonstrating broader applicability across more recent 3DVG baselines would strengthen the generality of our framework. In our revision, we have extended the Refer-Judge experiment to include ConcreteNet.
>
> Due to the relatively heavy architecture of ConcreteNet, reproducing full training following the official configuration required approximately 4–5 days. As of submission, we have completed 235/400 training epochs. Encouragingly, we already observe an improvement of +0.55 in Overall Acc@0.5 over the original ConcreteNet, and we expect further gains once full training is complete. These updated results will be included in our revised manuscript.
>
> We are confident that completing the training will yield further gains, and we will include the results in the revised version of the paper.
>
> **Part 4: Table 2b Interpretation**
>
> Thank you for highlighting this. This observation indeed reveals a critical property we discovered during the evaluation of Refer-Judge's impact on baseline models.
>
> As reported in Table 2b and further elaborated in Appendix C (Table 7), baseline models often exhibit higher performance on toxic data, even though the referring expressions in those samples may be ambiguous, paradoxical, or logically invalid. For instance, at a toxic threshold of ≤2, the baseline model achieves +2.29% higher Acc@0.5.
>
> We believe this phenomenon stems from implicit overfitting to annotation artifacts in flawed samples, where the model may learn dataset-specific priors rather than generalizable grounding capabilities. This raises concerns about evaluation reliability when training and test data contain similar toxic patterns.
>
> Furthermore, the experiments in Appendix C show that as the toxic threshold becomes more inclusive (i.e., including moderately flawed samples), Refer-Judge-filtered models achieve increasingly stronger gains (e.g., +2.16% at threshold ≤4), reinforcing the benefit of data purification.
>
> **Q5: Ambiguity in Dataset Filtration Process**
>
> Thank you for pointing this out. We acknowledge that our original explanation of the positive set and the misleading set construction may have been unclear. We clarify that these sets are not derived via MLLM predictions or heuristic rules.
>
> Specifically, for a given referring expression in ScanRefer, which targets a unique ground-truth object, we identify ScanNet frames in which this object is visible, constituting the positive set. In contrast, the misleading set consists of frames from the same scene that contain objects of the same semantic category but do not include the ground-truth target object. These misleading views simulate plausible yet incorrect matches and are essential for enabling ambiguity-aware reasoning.
>
> We will revise the manuscript to clarify this preprocessing process in detail.
>
> **Q6: Computation and Cost Concerns**
>
> Thank you for raising this important point. We clarify that the actual cost of using Refer-Judge for dataset auditing is entirely affordable and practical.
>
> As reported in Table 5, a full Refer-Judge evaluation pass over the ScanRefer-Justice dataset consumes approximately 112.8M tokens (97.16M input, 15.64M output). Based on the current GPT-4o API pricing (USD 1.25/M input tokens, USD 5/M output tokens), this results in a total cost of 200 USD. Scaling to the full ScanRefer dataset, the total estimated cost would be 2,500 USD.
> More importantly, we also evaluated a cost-efficient variant of our system using GPT-4.1-mini, which completes the Refer-Judge process over ScanRefer-Justice for 32 USD. This highlights that Refer-Judge is highly scalable, modular, and not tied to expensive APIs.
>
> Compared to manual annotation, which required over 200 human hours for 3,001 samples (with an equivalent market cost likely exceeding several thousand USD), our approach offers a significantly more cost-effective and labor-saving alternative.
>
> **Closing Remarks**
>
> We appreciate the reviewer’s thoughtful critique. While we acknowledge our method is not architecturally novel, we believe Refer-Judge offers a practical and extensible solution to a critical and underexplored problem in dataset reliability. We respectfully ask the reviewer to reconsider their evaluation in light of these clarifications and updates.

---

### Official Review · Reviewer_Udtb · 2025-07-05

**Clarity:** 2
**Significance:** 2
**Originality:** 2
**Rating:** 4
**Confidence:** 2

**Summary:**

This paper addresses the issue of toxic data in 3D Visual Grounding (3DVG) datasets, which contain paradoxical or ambiguous annotations that hinder model performance. The authors propose Refer-Judge, a framework inspired by the judicial system’s deliberative process, using Multimodal Large Language Models (MLLMs) to detect and filter such toxic data.

The framework employs a Jury-and-Judge Chain-of-Thought paradigm:
- Jurors evaluate data from four perspectives (Logic, Consistency, Distinguishability, Ambiguity), identifying logical contradictions and ambiguities.
- Judges refine these evaluations using a Corroborative Refinement strategy, reorganizing visual/textual evidence to resolve uncertainties.

Experiments on the ScanRefer-Justice dataset show Refer-Judge achieves human-level discrimination and improves baseline model performance when toxic data is removed. The core contribution is a principled, multi-perspective reasoning system for 3DVG data quality assessment.

**Questions:**

- Reproducibility Crisis: Why does the paper omit code for the "stitching operation" and Jury-Judge prompting templates? Without these, results cannot be verified.
- Biased Dataset Analysis: How does Refer-Judge perform on ScanRefer's original test set, which has a different complexity distribution? The reported gains on ScanRefer-Justice may not generalize.
- Cost-Effectiveness Failure: A $10,000 budget for GPT-4o API calls is prohibitive for most labs. Why not compare against cheaper alternatives (e.g., Llama-3.2 with fine-tuning)?.
- Theoretical Vacuity: What is the formal relationship between the judicial analogy and the proposed reasoning stages? The metaphor lacks predictive power for model design.

**Ethical Concerns:**

["NO or VERY MINOR ethics concerns only"]

**Final Justification:**

Based on the reviews and rebuttals, i've updated my review to boarderline accept.

**Limitations:**

- Computational Infeasibility: The paper downplays high costs, failing to compare against lightweight methods (e.g., rule-based filters) for toxic data detection.
- Dataset Bias: ScanRefer-Justice overrepresents complex scenes (20% complex samples vs. 6.3% in ScanRefer), skewing performance metrics.
- MLLM Dependency: Performance collapses with open-source LLMs (e.g., LLAMA-3.2 achieves 67.88% agreement), yet the paper frames GPT-4o results as a universal solution.

**Quality:**

2

**Strengths And Weaknesses:**

Strengths
- Original Framework Design: The judicial-inspired Jury-Judge paradigm is novel, providing structured multi-perspective reasoning absent in prior work. This decomposes complex scene-level evaluation into manageable dimensions.
- Comprehensive Evaluation: By integrating logical, consistency, and visual distinguishability checks, Refer-Judge addresses multiple sources of toxic data (e.g., internal contradictions, referential ambiguity).
- Empirical Validation: Experiments on ScanRefer-Justice and ScanRefer demonstrate significant improvements in data purification and model performance, with GPT-4o achieving near-human accuracy.
- Practical Impact: The framework directly addresses a critical challenge in 3DVG, where dataset quality has been overlooked. Filtering toxic data leads to tangible gains in downstream tasks.

Weakness
- Lack of Theoretical Innovation: The judicial analogy is superficial, merely renaming standard multi-agent reasoning (jurors) and refinement (judges). Concepts like "twin thought" replicate existing CoT strategies (e.g., parallel reasoning in GoT [3]) without novel theoretical grounding.
- Inadequate Evaluation Scope: Experiments exclusively use ScanRefer-Justice, a dataset curated to emphasize complex scenes.
- Performance on real-world 3DVG datasets (e.g., SUNCG) or outdoor environments is untested, undermining generalizability.
- Questionable Computational Efficiency: The framework incurs exorbitant costs ($10,000 in API fees) for marginal gains (e.g., +1.6% Acc@0.5 for 3DVLP). This makes it impractical for large-scale datasets, a critical limitation unaddressed in the paper.
- Flawed Baseline Comparisons: Baseline models (e.g., ScanRefer, 3DVLP) are evaluated on purified data without controlling for data splitting biases. The reported improvements may stem from dataset curation rather than the framework itself.
- Ambiguous Refinement Mechanisms: The Corroborative Refinement strategy lacks clear definitions. For example, "stitching operation" (sti·) is undefined, and the replacement of evaluations (replace()) lacks transparency, making reproducibility impossible.

---

> ### Author Rebuttal · Authors · 2025-07-31
>
> We sincerely thank the reviewer for the detailed feedback and the opportunity to clarify and strengthen our submission. Below, we address each of the raised concerns.
>
> **Q1: Theoretical framing and judicial analogy**
>
> We respectfully disagree that the proposed framework merely renames standard reasoning mechanisms. While Refer-Judge is rooted in CoT-based LLM reasoning, it is, to the best of our knowledge, the first agentic, deliberation-based system aimed specifically at evaluating complex dataset quality in 3DVG. We believe this Jury-and-Judge CoT design represents a structured and extensible paradigm for LLM-as-a-Judge applications. Its value lies in a novel and structured formulation of LLM-based data auditing—a dimension increasingly relevant in the era of foundation models.
>
> We appreciate prior efforts such as GoT, which explore topological variations of CoT graphs. However, as we stated in our related works, *most CoT-based methods are tailored for narrow-scope tasks, including arithmetic reasoning (like GoT) or short image-text QA (like CoCoT, DCoT)*. These methods demonstrate the existence of complex CoT topologies, but not their utility in executing open-ended, multi-perspective tasks like data auditing in 3DVG.
>
> Our framework introduces:
>
> - Modular Role-Driven Reasoning: Jurors are semantically differentiated agents, each targeting a key perspective: logical, consistency, distinguishability, and ambiguity. This design goes beyond ensemble prompting, aligning with how real-world juries solicit diverse perspectives. Unlike GoT-style parallel agents, Refer-Judge enforces cross-perspective reasoning and inter-agent role separation.
> - Corroborative Refinement and Decision Aggregation: The judge layer is not a majority vote but a structured aggregation then re-evaluation mechanism, including conflicts recognition, information reorganizing, and corroborative scoring. This mitigates the effects of unreliable observations. As shown in Table 4, our refinement significantly outperforms both GoT and CoT-SC, underscoring its effectiveness.
> - Twin Thought Reuse Across Roles: While structurally parallel, our Twin Thought concept reflects multi-role reuse of reasoning traces. For example, existence judgments are shared across J_a and J_d but differ in their inputs (positive vs. misleading view sets), resulting in divergent semantic functions.
>
> While we acknowledge that Refer-Judge is not focused on in-depth exploration of CoT topology, we believe this framework represents methodological innovations in LLM-as-a-Judge field.
>
> **Q2: Evaluation Scope and Generalizability (Dataset Bias)**
>
> We acknowledge that ScanRefer-Justice emphasizes more complex and ambiguous cases, and we explicitly stated this design choice in the paper. Our goal was to create a stress-test benchmark that evaluates grounding robustness under challenging conditions.
>
> As noted in Appendix A.2, manual validation of toxic samples requires significantly more effort than initial annotation. Consequently, it was infeasible for us to manually validate the full ScanRefer dataset. Even for the ScanRefer-Justice subset, five trained annotators spent over 200 human-hours.
>
> Moreover, we believe that concentrating on these “pressure samples” is critical for revealing the limits of current MLLMs and understanding the comparative strengths of different Refer-Judge components. Particularly in scenarios where full human verification is impractical, ensuring sufficient coverage of complex edge cases is more impactful than uniformly sampling the entire dataset.
>
> **Q3: Expensive Computational cost**
>
> Thank you for raising the cost concern. We would like to clarify that the $10,000 figure reflects our cumulative expenditure across all stages during the writing and experimentation phase. The actual cost of running Refer-Judge for data filtering is significantly lower and well within a practical range.
>
> As shown in Table 5, a complete Refer-Judge pass over the ScanRefer-Justice dataset involves approximately 112.8M tokens (97.16M input, 15.64M output). Based on current API pricing  (USD 1.25/M input, USD 5/M output), this equates to a total cost of 200 USD. Scaling up to the full ScanRefer dataset, the estimated cost would be 2,500 USD, for the original dataset contains simpler cases, which require fewer visual and ambiguity evaluations.
>
> Furthermore, as demonstrated in Table 1, substituting GPT-4o with GPT-4.1-mini achieves comparable performance at a much lower cost, ~$32 for the ScanRefer-Justice dataset. This highlights the scalability and adaptability of Refer-Judge.
>
> Compared to manual annotation or the development of handcrafted rules for scene-specific ambiguity detection, we believe Refer-Judge offers a more cost-effective, reusable, and modular solution for dataset purification in 3DVG and beyond.
>
> **Q4: Baseline comparison and dataset split bias**
>
> We clarify that in Table 2(a) and Table 6, both ScanRefer and 3DVLP models are evaluated on the original ScanRefer test set, using identical train/val/test splits for fair comparison. This ensures our data purification impacts training quality.
>
> At the same time, we also clearly explained in the original manuscript that the experiments in Table 2 (b) and Table 7 further evaluated the performance of the model on toxic data and purified data. As we mentioned in the response of Q4, this move was to analyze whether the model would overfit the toxic samples.
>
> **Q5: Ambiguous Mechanism Descriptions and Reproducibility**
>
> We appreciate the reviewer’s comment regarding the clarity of terminology and operations in the manuscript. We acknowledge that the “stitching operation” and “replacement operation” were insufficiently explained in the original text, and we will provide clearer definitions in the revised version.
>
> Concretely, the stitching operation refers to a simple visual composition procedure applied to Corroborative Pairs: two images, one being the suspicious view and the other the assistance view. These two images are horizontally concatenated with aligned edges, as illustrated by the thumbnail in Figure 3, to form a richer and more context-aware visual prompt. This stitched image provides the judge with a more complete reference for final assessment.
>
> The replacement operation substitutes the original (possibly erroneous or low-confidence) score and rationale produced by individual jurors with the refined outputs from the Judge module, which integrates broader evidence and resolves conflicts. This enables more accurate and globally consistent evaluations.
>
> Regarding reproducibility, we note that the main paper presents a simplified view of the prompting templates and data flow operations for the sake of clarity. However, to support reproducibility, we have included all implementation details in the supplementary materials. These include:
>
> - Preprocessing scripts for both ScanRefer and ScanRefer-Justice
> - The end-to-end pipeline for Refer-Judge
> - The used prompting templates
> - The ScanRefer-Justice benchmark
>
> We hope this clarifies the implementation of our proposed method.
>
> **Q6: MLLM Dependency and Lack of Fine-Tuning on Open-Source LLMs**
>
> Indeed, performance varies by model. LLaMA-3.2 achieves lower agreement, showing current limitations of open models. However, Refer-Judge remains functional across all tested LLMs. The modular design and corroboration mechanisms mitigate individual hallucinations, as shown in Appendix D.
>
> As for fine-tuning LLaMA or other open models, we considered this, but such tuning requires substantial task-specific annotations (e.g., annotating ambiguity types, visual correspondence under multiple views, and instance-level scene justifications). This would require significant annotation effort, defeating the purpose of scalable data auditing.
>
> Moreover, as stated in Q3, cost is already manageable with GPT-4.1-mini, which offers excellent cost-performance trade-offs.
>
> **Q7: Broader Evaluation and Generalization**
>
> We thank the reviewer for highlighting the importance of broader evaluation. We agree that validation beyond a single dataset improves the perceived generalizability of Refer-Judge.
> In response to your suggestion and that of Reviewer # KTYR in limited time, we first conducted a small-scale study on the Nr3D dataset, which provides viewpoint splits and distractor-based setups. Manual review by human experts confirmed that ambiguous and paradoxical expressions also exist in Nr3D.
>
> Encouragingly, Refer-Judge aligned well with expert judgment in these cases. For example, for sample #23273 (*The brown arm chair below the painting close to the purple couch*), although the original annotation marked the listener’s pick as correct, the scene (scene0520_00) actually contains two valid armchairs below different paintings, making the expression ambiguous. Refer-Judge successfully identified this.
>
> Interestingly, we also observed cases where annotations were marked as failed in Nr3D (due to speaker-listener mismatch) but turned out to be valid on re-check. For instance, sample #39432 (*It is a shelf with a poster on it touching the corner of the room behind the green chair*), though judged incorrect in Nr3D, can be clearly grounded to a unique shelf (seen in 000500.jpg of scene0565_00) by both experts and Refer-Judge.
>
> We will include these findings in the revised version, and we welcome future work extending Refer-Judge to other datasets such as Sr3D and SUNCG.
>
> **Closing remarks**
>
> We thank the reviewer again for their rigorous critique. We will revise the paper to clarify system design, expand evaluation scope, and emphasize reproducibility and cost-efficiency. While Refer-Judge may not introduce new theoretical formalisms, we believe it makes a methodologically novel contribution to dataset quality assessment in 3DVG, offering an effective, interpretable, and modular solution to a widely underexplored challenge. We respectfully ask the reviewer to reconsider the overall evaluation.

---

### Official Review · Reviewer_63DY · 2025-07-05

**Clarity:** 2
**Significance:** 2
**Originality:** 2
**Rating:** 4
**Confidence:** 3

**Summary:**

This paper introduces a novel  Refer-Judge framework designed to remove ambiguous or paradoxical annotations in 3D visual grounding (3DVG) datasets. Inspired by judicial deliberation, multiple jurors analyze each sample from four complementary perspectives, and their outputs are then refined and consolidated by a panel of judges. Experiments on the ScanRefer-Justice dataset show that Refer-Judge achieves human-comparable performance in detecting problematic samples and that filtering toxic data leads to improvements in downstream 3DVG tasks.

**Questions:**

1. The author should discuss the relation between computation cost (e.g., by token or flops) and the refinement gain, which is critical for the scalability of the proposed method.
2. Potential bias analysis would also be helpful since existing MLLMs exhibit a lot of hallucination.

**Ethical Concerns:**

["NO or VERY MINOR ethics concerns only"]

**Final Justification:**

I appreciate the authors' effort and detail response. Thanks for pointing out that GPT-4.1-mini could achieve comparable results with lower price. The performance analysis is reasonable and I will raise my score.

**Limitations:**

yes

**Quality:**

2

**Strengths And Weaknesses:**

Strength

1. The method closely mimics human deliberative processes and yields reasonable performance improvement.
2. The ablation study comprehensively demonstrates that the designed modules are important.

Weakness
1. The proposed framework involves a large amount of computation by querying strong and expensive MLLMs multiple times, while the performance improvement on downstream tasks seems limited.
2. The quality of refinement seems to largely rely on the MLLM choice, while there is little discussion of failure cases or model bias.

---

> ### Author Rebuttal · Authors · 2025-07-31
>
> We sincerely thank the reviewer for the thoughtful feedback and detailed comments. Below, we address your concerns regarding scalability, computational cost, and model dependency.
>
> **Q1: Computation cost vs. refinement gain (scalability concern)**
>
> We would like to clarify a potential misunderstanding regarding computational cost. While Refer-Judge involves multiple structured MLLM queries per scene, its overall cost remains manageable and significantly lower than human annotation for comparable tasks.
>
> As shown in Table 5 (ID 4), a full Refer-Judge pass on the ScanRefer-Justice dataset (3,001 samples) consumes approximately 112.8M tokens, with 97.16M input tokens and 15.64M output tokens. Based on current GPT-4o pricing (USD 1.25/M input, USD 5/M output), this translates to 200 USD total. Scaling up to the full ScanRefer dataset (51,583 expressions), the estimated cost would be 2,500 USD, as the original dataset contains simpler cases, which require fewer visual and ambiguity evaluations.
>
> Importantly, this cost is a one-time, offline expenditure. Refer-Judge is designed for data curation or pretraining supervision, where runtime latency is less critical than quality assurance. In contrast, human annotation of ScanRefer-Justice required over 200 human-hours by 5 trained annotators, which is not only more expensive, but also less scalable or reproducible.
>
> To further reduce the running cost of Refer-Judge, we also recommend using a more cost-effective MLLM. As we reported in Table 1, using GPT-4.1-mini as a replacement also produces comparable results, and the inference overhead on the ScanRefer-Justice dataset can be further reduced to a negligible $32, demonstrating the scalability potential of our work.
>
> Meanwhile, the discussion on the correlation between computation cost and the quality of toxic data evaluation has been fully discussed in Appendix C. The ablation experiment in Table 8 clearly shows the impact of each component in Refer-Judge on token cost and its correlation with the refinement gain.
>
> We hope this clarifies that Refer-Judge is both practically affordable and structurally scalable for real-world deployment in offline data refinement.
>
> **Q2: Limited improvement on downstream tasks**
>
> We respectfully clarify that Refer-Judge leads to consistent and non-trivial improvements on downstream 3DVG models, without modifying their architectures. As shown in Table 2(a), filtering the training data with Refer-Judge improves Overall Acc@0.5 by +1.43% (ScanRefer) and +1.6% (3DVLP).
>
> Additionally, we believe that baseline models trained on the original ScanRefer dataset may implicitly learn annotation priors embedded in toxic samples. In cases involving paradoxes or ambiguities, human annotators may exhibit consistent biases in selecting certain objects as ground truth (e.g., favoring closer or more salient targets). This leads to models overfitting to these spurious but learnable annotation patterns, rather than truly reasoning over spatial and semantic consistency.
>
> Specifically, our analysis in Appendix C (Table 7) reveals that baseline models may implicitly overfit to annotation biases in toxic samples. For example, the baseline model outperforms 3DVLP+Refer-Judge on toxic data, even though the referring expressions in those samples may be ambiguous, paradoxical, or logically invalid. For example, at a toxic threshold of ≤2, the baseline even achieves +2.29% higher Acc@0.5 on toxic samples, indicating inflated scores due to spurious correlations. In contrast, Refer-Judge-filtered models achieve more reliable performance on valid data and show greater gains as the toxic threshold expands (e.g., +2.16% at threshold ≤4), further supporting the value of data purification.
>
> We would also like to emphasize that data quality remains the cornerstone of deep learning. While we are deeply grateful for foundational contributions such as ScanRefer in advancing 3D visual grounding, we believe that identifying and addressing latent flaws in these datasets is equally essential for the long-term progress of 3DVG and spatial intelligence.
>
> As stated in our conclusion, “*We hope this work offers a new lens on dataset reliability in 3D scene understanding and serves as a foundation for more trustworthy training and evaluation practices.*” Beyond improving model performance, our goal is to spark broader reflection on the reliability of mainstream 3DVG datasets, especially given the inherent challenges of annotation in such complex, multimodal settings.
>
> **Q3: Model bias and failure case discussion**
>
> Thank you for raising this important concern.
>
> To evaluate the generalizability and robustness of the Refer-Judge framework, we tested it using a diverse set of MLLMs, including GPT-4o, GPT-4.1-mini, Gemini 2.5, Grok-3, and LLaMA-3.2 (11B). These models differ significantly in architecture, training corpus, and reasoning capabilities. Naturally, we observed performance variance across models. For example, LLaMA-3.2, a much smaller open-source model, achieved a lower agreement score (67.88%), underscoring the difficulty of toxic data identification in complex, multi-view 3D settings.
>
> However, we would like to clarify that these discrepancies reflect model capability gaps, rather than indicating a lack of robustness in the Refer-Judge framework itself. In fact, the component ablation experiments based on GPT-4o (see Table 8) consistently demonstrate the contribution of each module to final prediction quality. We welcome future work that leverages our proposed ScanRefer-Justice benchmark to further evaluate and enhance these findings across broader model families.
>
> On the other hand, we acknowledge that our current manuscript lacks a detailed discussion of failure cases, particularly those arising from MLLM hallucinations. While Appendix D offers brief visualizations for normal, hard, and complex scenes, the error analyses are not sufficiently comprehensive. In our revision, we will include a deeper case study section focusing on these issues.
>
> For example, in the reported *normal case*, the uniqueness refinement module hallucinated that a TV *"fails to match the specified `square` shape,"* which distorted the final judgment. This failure is traceable in the District Judge’s score breakdown, where the corresponding distinguishability rank was marked as [10: 2]. Nevertheless, because the other visual views were accurately interpreted, the final aggregated confidence score remained high, demonstrating Refer-Judge's resilience to localized hallucinations.
>
> We will also correct the mismatched image ordering in the previously reported case studies to avoid confusion.
>
>
> **Closing remarks**
>
> We thank the reviewer again for raising these important points. We hope our clarifications and additional analysis have addressed your concerns and demonstrated the practicality, scalability, and reliability of Refer-Judge. We respectfully ask the reviewer to reconsider the overall evaluation.

---

### Official Review · Reviewer_hBqW · 2025-07-07

**Clarity:** 3
**Significance:** 3
**Originality:** 3
**Rating:** 5
**Confidence:** 4

**Summary:**

This paper proposes Refer-Judge, an agentic system to detect toxic data in 3D visual grounding. The framework employs four jurors to assess different aspects of the data quality in 3DVG, followed by a judge refinement and aggregation procedure.
The proposed framework achieves an accurate assessment that is close to human performance.
Experiments show existing 3DVG approaches can benefit from fine-tuning using the purified data.

**Questions:**

- What is the typical process time of one scene?

**Ethical Concerns:**

["NO or VERY MINOR ethics concerns only"]

**Final Justification:**

The authors addressed my questions about the baseline, details of the misleading set, and the running time.

**Limitations:**

yes

**Quality:**

4

**Strengths And Weaknesses:**

Strength:
- The idea and the framework design are insightful.
- The paper is well-written and easy to follow.
- The experiments are comprehensive.

Weakness:
- In Tab. 2 (a), the performances of ScanRefer [1] and 3DVLP [2] show impressive performance gains after using the Refer-Judge framework. However, it will be more persuasive if the framework can consistently boost at least one more recent work (given ScanRefer [1] was published in 2020).
- How is the misleading view set $I_{mis}$ described in L. 188 constructed.

[1] Chen, Dave Zhenyu, Angel X. Chang, and Matthias Nießner. "Scanrefer: 3d object localization in rgb-d scans using natural language." European conference on computer vision. Cham: Springer International Publishing, 2020.
[2] Zhang, Taolin, et al. "Vision-language pre-training with object contrastive learning for 3d scene understanding." Proceedings of the AAAI Conference on Artificial Intelligence. Vol. 38. No. 7. 2024.

---

> ### Author Rebuttal · Authors · 2025-07-31
>
> We thank the reviewer for the encouraging comments and recognition of the insights, clarity, and comprehensiveness of our work. We appreciate the helpful suggestions and provide detailed responses below.
>
> **Q1: Additional recent baselines beyond ScanRefer/3DVLP (Tab. 2(a))**
>
> We agree with the reviewer that demonstrating broader applicability across more recent 3DVG baselines would strengthen the generality of our framework. In our revision, we have extended the Refer-Judge experiment to include ConcreteNet [ECCV 2024], a recently published and strong 3DVG baseline.
>
> Due to the relatively heavy architecture of ConcreteNet, reproducing full training following the official configuration required approximately 4–5 days on our hardware setup. As of submission, we have completed 235 out of 400 training epochs on our Refer-Judge-purified ScanRefer dataset. Encouragingly, we already observe an improvement of +0.55 in Overall Acc@0.5 over the original ConcreteNet baseline, and we expect further gains once full training is complete. These updated results will be included in our revised manuscript.
>
> In addition, we emphasize that our original choice of ScanRefer and 3DVLP as testbeds was deliberate. These models represent different levels of model complexity and data fitting capacity, allowing us to assess whether data toxic impacts both lightweight and powerful 3DVG systems. The consistent performance gains across these diverse architectures support our broader claim: toxic data negatively affects a wide range of 3DVG models, making data auditing an essential step in trustworthy model training.
>
> **Q2: Construction of the misleading view set (Line 188)**
>
> Thank you for pointing out this potential ambiguity. We realize our description of the positive set and the misleading set construction may not have been sufficiently clear in the original manuscript.
>
> To clarify, the generation of these two image sets is grounded in the existing annotations of the ScanRefer and ScanNet datasets. Specifically:
>
> - For each referring expression in ScanRefer, the corresponding ground truth object is identified. Leveraging the alignment between ScanNet's RGB frames and object-level 3D annotations, we determine which images in the scene contain visible projections of the target object. These images comprise the positive set, ensuring that each view indeed depicts the intended referent.
> - In contrast, the misleading set is formed by selecting images from the same scene that do not contain the ground truth object, but do contain other instances of the same object category. These views are intentionally constructed to be partially correct yet referentially incorrect, serving as distractors that may tempt a model to violate exclusivity constraints.
>
> This construction protocol ensures that images in the I_mis set provide both visual and semantic ambiguity, which is essential for evaluating whether the referring expression is uniquely resolvable.
>
> We will revise the manuscript to clarify this process in more detail.
>
> **Q3: Inference time per scene**
>
> We appreciate the reviewer’s interest in the runtime characteristics of Refer-Judge. However, we note that per-scene inference time is not a meaningful performance bottleneck in our setting, as the framework is designed for offline data curation, not real-time inference.
>
> In practice, thanks to the independent nature of each sample’s evaluation, Refer-Judge naturally supports batch-mode parallelization. With a 32-thread setup, the average end-to-end time (including all four juror evaluations and judge refinement) is approximately 6 seconds per sample.
>
> Given that Refer-Judge is intended for one-time dataset purification, whether for improving 3DVG training reliability or constructing instruction-following benchmarks for MLLMs, throughput is not a limiting factor, and quality is prioritized over latency.
>
> **Closing remarks**
>
> We hope these clarifications and additional experiments address your concerns. Thank you again for your constructive and supportive review.

---

> > ### Comment · Reviewer_hBqW · 2025-08-06
> >
> > Dear Authors,
> >
> > Thanks for your reply. My questions have been addressed.
> >
> > Best Regards,
> > The Reviewer

---

### Decision · Program_Chairs · 2025-09-17

**Decision:**

Accept (poster)

**Comment:**

This paper presents Refer-Judge, a Jury-and-Judge Chain-of-Thought framework that leverages MLLMs to uncover toxic data in 3D visual grounding. The approach is conceptually novel, framing data auditing as a structured deliberative process, and shows human-level discrimination on ScanRefer-Justice along with measurable gains for downstream baselines. Reviewer hBqW found the idea insightful and the experiments comprehensive, recommending acceptance. Reviewer 63DY initially raised concerns about computational overhead and MLLM dependency, but after rebuttal and evidence of affordable cost-effective variants (e.g., GPT-4.1-mini), raised their score. Reviewer Udtb and KTYR emphasized limited methodological novelty and dataset scope, but the rebuttal provided clarifications, additional results on ConcreteNet and Nr3D, and clearer explanations of the filtration process. While not algorithmically deep, the work makes a timely and important contribution to improving dataset reliability in 3D vision-language tasks. Given the strong motivation, effective rebuttal, and potential impact, I recommend acceptance.